# An allele-selective inter-chromosomal protein bridge supports monogenic antigen expression in the African trypanosome

Joana R. C. Faria ●[1,2,3] ✉, Michele Tinti ●[1], Catarina A. Marques ●[1,6], Martin Zoltner ●[1,7], Harunori Yoshikawa ●[4,8], Mark C. Field[1,5] & David Horn ●[1] ✉

UPF1-like helicases play roles in telomeric heterochromatin formation and X-chromosome inactivation, and also in monogenic variant surface glycoprotein (VSG) expression via VSG exclusion-factor-2 (VEX2), a UPF1-related protein in the African trypanosome. We show that VEX2 associates with chromatin specifically at the single active *VSG* expression site on chromosome 6, forming an allele-selective connection, via VEX1, to the *trans*-splicing locus on chromosome 9, physically bridging two chromosomes and the *VSG* transcription and splicing compartments. We further show that the VEX-complex is multimeric and self-regulates turnover to tightly control its abundance. Using single cell transcriptomics following VEX2-depletion, we observed simultaneous derepression of many other telomeric *VSG*s and multi-allelic *VSG* expression in individual cells. Thus, an allele-selective, inter-chromosomal, and self-limiting VEX1-2 bridge supports monogenic *VSG* expression and multi-allelic *VSG* exclusion.

Mechanisms underpinning monogenic, or allele-selective, expression persist as major outstanding mysteries in eukaryotic biology[1]. Striking examples include the expression of a single olfactory receptor in odorant sensory neurons, and X-chromosome inactivation in female mammals[2]. Monogenic expression also operates in protozoal pathogens, such as *Plasmodium falciparum* and *Trypanosoma brucei*, which undergo surface antigenic variation to evade host immune responses[3–5]. These latter examples typically involve the stochastic selection of a single allele from a large family. Despite decades of intense study, the molecular mechanisms facilitating the selection and maintenance of expression of a single active allele, coordinated with the silencing of all others, and stochastic, low-frequency transcriptional switching, remain poorly understood.

African trypanosomes are human and veterinary pathogens that are transmitted between mammalian hosts by tsetse flies. These parasites remain an exemplar organism for studies on monogenic expression and immune evasion. Indeed, *T. brucei* are 'masters of disguise', relying on ~2600 variant surface glycoprotein genes and pseudogenes[6] for antigenic switching and to sustain persistent infections[7]. Despite this vast genetic repertoire, *VSG* genes are exclusively expressed from one of ~15 sub-telomeric, polycistronic transcription units, designated *VSG* expression sites (*VSG*-ESs)[8]. RNA polymerase-I transcribes the single active *VSG*-ES within an expression-site body (ESB), a sub-nuclear and extranucleolar transcription factory[9], while other *VSG*-ESs, despite sharing a common set of DNA elements, are transcriptionally repressed.

Inter-chromosomal interactions between promoters, enhancers and super-enhancers are implicated in selective gene expression, such as the interactions at the heart of olfactory receptor monogenic expression[10–12]. The active *VSG* locus and a *trans*-splicing locus involved

[1]Wellcome Centre for Anti-Infectives Research, Biological Chemistry and Drug Discovery, School of Life Sciences, University of Dundee, Dundee, UK. [2]Biology Department, University of York, York, UK. [3]York Biomedical Research Institute, University of York, York, UK. [4]Gene Regulation and Expression, School of Life Sciences, University of Dundee, Dundee, UK. [5]Biology Centre, Czech Academy of Sciences, Institute of Parasitology, České Budějovice, Czech Republic. [6]Present address: Wellcome Centre for Integrative Parasitology, University of Glasgow, Glasgow, UK. [7]Present address: Faculty of Science, Charles University in Prague, Biocev, Vestec, Czech Republic. [8]Present address: Division of Cell Signaling, Fujii Memorial Institute of Medical Sciences, Institute of Advanced Medical Sciences, Tokushima University, Tokushima, Japan. ✉e-mail: joana.correiafaria@york.ac.uk; d.horn@dundee.ac.uk

in RNA maturation in trypanosomes are also proximal, spatially integrating both transcription and splicing to enhance VSG expression[13], facilitating the production of ~10% of all cellular mRNA from a single active *VSG* gene[9,13–15]. Indeed, proximity to nuclear condensates composed of RNA processing factors can increase gene expression in both mammals[16,17] and trypanosomes[13,18].

VSG exclusion factor 2 (VEX2)[19] associates with VEX1 in a transcription-dependent manner within a *VSG* transcription and RNA *trans*-splicing subcompartment, orchestrating both single allele choice and the exclusion of other *VSGs*[13,19–21]. Here, we show that VEX2 is a *VSG* allele-selective and self-limiting factor that, with VEX1, forms a physical bipartite inter-chromosomal bridge to support multi-allelic exclusion.

## Results
### A bridge between two chromosomes
VEX2 co-localises with the hemizygous active *VSG* expression site (*VSG*-ES), on chromosome 6 in the strain analysed here, while VEX1 co-localises with the diploid 'spliced leader' *SL*-array on chromosome 9[13]. VEX1 ChIP-Seq previously revealed interactions with the telomeric ends of multiple *VSG*-ESs[13,19]. To explore VEX2-chromatin interactions, we performed ChIP-Seq with affinity-purified VEX2 (Fig. 1a–e, Supplementary Figs. 1 and 2 and Supplementary Data 1, sheets 1–2). Genome-wide peak calling revealed a remarkably specific association with only one *VSG*-ES, the active site, at one end of one copy of chromosome 6 (Fig. 1a–c and Supplementary Fig. 2b, c). A circos plot reveals this signal specifically in the strain expressing affinity-tagged VEX2 and specifically at the active *VSG*-ES (BES1) in that strain (Fig. 1a); note that BES1 has been mapped to chromosome 6[6] but because genome sequencing projects for *T. brucei* have not yet delivered telomere-to-telomere chromosome assemblies, the telomeric *VSG*-ESs[8,22,23] are shown separately in the circos plot. VEX2 is associated with the full polycistronically transcribed region across this ES (Fig. 1b and Supplementary Fig. 2b, c), a distinct pattern relative to that of VEX1 (Fig. 1c), determined previously[19]. In contrast, we observed only moderate enrichment of the *SL*-array on chromosome 9 following VEX2 ChIP-Seq (Fig. 1a, d), consistent with an indirect interaction via VEX1; indeed, the *SL*-array was strongly enriched following VEX1 ChIP-Seq (Fig. 1d), as determined previously[13]. We also mapped the VEX2 ChIP-Seq data to a second genome assembly[6] that includes hemizygous sub-telomeric sequences (Supplementary Fig. 2d). As above, this circos plot reveals VEX2 chromatin binding specifically in the strain expressing affinity-tagged VEX2 and specifically at the active *VSG*-ES in that strain. To visualise the VEX2-chromatin association, we performed DNA-FISH using probes targeting the 50-bp repeats, which mark the *VSG*-ESs; and the *SL*-arrays, immunostaining VEX2[myc] in parallel. This analysis revealed VEX2 at the interface between a *VSG*-ES focus and one of the two diploid *SL*-arrays (Fig. 1f).

Having established that VEX2 binds the active *VSG*-ES, and having previously determined that VEX1 interacts with VEX2[19], we investigated these interactions further using super-resolution 3D-SIM (Structured Illumination Microscopy) (Fig. 2). Since VEX2 is a large protein (~224 kDa), we attempted to visualise the relative localisation of the termini by tagging both ends. Both ends were distinguishable using 3D-SIM (Fig. 2a), suggesting that a pool of VEX2 protein displays a common orientation at this site. We obtained similar results when the distinct tags were swapped to opposite ends of VEX2 (Fig. 2b, c). Next, we investigated the expression-site body (ESB) using Pol-I staining, or VEX1 positioning, relative to VEX2 N- or C-terminal tags (Fig. 2d–g). [GFP]VEX2 overlaps with VEX1[myc] in 57 ± 10% of G1 cells, decreasing to 32 ± 4% in S-phase, when VEX1 and VEX2 are more often adjacent (50 ± 7%) (Fig. 2d). In contrast, VEX2[GFP] overlaps with VEX1[myc] in a substantially higher proportion of G1 nuclei (81 ± 4%), again decreasing, to 57 ± 2%, in S phase (Fig. 2e). [myc]VEX2 co-localises with Pol-I at the ESB in 96 ± 2% of G1 nuclei, while VEX2[myc] co-localises with Pol-I at the

ESB in 86 ± 2% of G1 nuclei (Fig. 2f, g), associations that are moderately diminished during S-phase (Fig. 2f, g).

To investigate the role of *SL*-RNA in VEX2 sequestration, we knocked down Spliced Leader Array Protein 1 (SLAP1, Tb927.11.9950), which localises to both *SL*-arrays and is required for *SL*-RNA production[18]. As expected, one of two SLAP1 compartments was observed in proximity to the VEX2 compartment (Supplementary Fig. 3a). SLAP1 knockdown led to a depletion of VEX2, a loss of VEX2 foci in 79 ± 9% of nuclei, and a loss of detectable extranucleolar Pol-I labelled ESBs in 44 ± 10% of nuclei, (Supplementary Fig. 3b–f). This is consistent with the view that VEX2 sequestration is dependent upon mRNA *trans*-splicing[13].

Taken together, these results indicated that the VEX2 N-terminus extends towards the extranucleolar Pol-I compartment, while the VEX2 C-terminus extends towards a VEX1 compartment. Since VEX1 is known to bind *SL*-RNA arrays[13], and to form a complex with VEX2[19], which is now known to bind the active *VSG*-ES (Fig. 1), we conclude that the VEX1·VEX2 complex bridges two chromosomes, linking the active *VSG*-ES to a major RNA splicing locus. Moreover, the SLAP1 splicing factor is required to maintain VEX2 sequestration and the VEX1 and VEX2 association is dynamic during DNA synthesis.

### The VEX1-2 bridge is self-limiting
We next sought further insights into VEX2 protein-protein interactions, using a cell line in which both *VEX2*-alleles were precision-tagged (Supplementary Fig. 4a)[24]. We affinity-purified [GFP]VEX2 following cryo-milling, which revealed an association with VEX1 and the three subunits of the CAF-1 (Chromatin Assembly Factor 1) histone chaperone (Fig. 3a, Supplementary Fig. 4b–d and Supplementary Data 1, sheets 3–4), consistent with prior analysis of affinity-purified VEX1[19]. The VEX1-VEX2 interaction was further demonstrated by reciprocal co-immunoprecipitation analysis (Supplementary Fig. 4e). Notably, the *T. brucei* telomere-binding proteins (TIF2, TRF, TelAP1 and Pol Theta) were also moderately enriched (Fig. 3a). The telomere-binding proteins were also enriched using an alternative buffer (Supplementary Fig. 4b), as were RNA polymerase-II associated factor 1 (PAF1) subunits[25] (Supplementary Fig. 4c) and a subset of nucleolar factors (Supplementary Fig. 4d). These findings are consistent with the telomeric location of the active *VSG* and its spatial proximity to a major splicing locus, within a prominent RNA Pol-II transcription compartment. One notable telomere-binding protein, RAP1[26,27], was not enriched, consistent with the view that RAP1 is primarily associated with chromatin at repressed *VSG*-ESs[28].

Next, to investigate how the VEX/CAF-1 complex assembles in native conditions and to further interrogate protein-protein interactions, we used native PAGE following knockdown of individual VEX complex components (Fig. 3b, c and Supplementary Fig. 5a–d). VEX1[myc] was predominantly present as oligomeric forms between ~600–1000 kDa, with the slower-migrating forms diminished following VEX2 depletion (Fig. 3b), and the faster-migrating forms diminished following CAF-1 depletion (Supplementary Fig. 5b), indicating that VEX1 associates with either VEX2 and/or CAF-1. A less abundant species migrating at ~240 kDa was also observed, consistent with a VEX1 dimer. Taken together, these results suggest that VEX1 dimers or higher oligomers interact with VEX2 and/or CAF-1. [myc]VEX2 monomers were detected migrating at ~240 kDa, while [myc]VEX2 was predominantly present as polydisperse oligomeric forms migrating between 500 and 1000 kDa, a distribution that was maintained following VEX1 or CAF-1 depletion (Fig. 3c and Supplementary Fig. 5d). These results suggest that a substantial pool of VEX2 acts independently of VEX1, also explaining why VEX1 or CAF-1b depletion previously failed to disrupt VEX2 foci[19]. CAF-1b is a subunit of a hetero-trimeric complex of 220 kDa (256 kDa including the myc-tag in CAF-1b[myc]; Supplementary Fig. 5a, c). Our results show that a pool of *T. brucei* CAF-1 associates with VEX1 (~500–720 kDa, Supplementary Fig. 5a), while we see relatively little VEX2/CAF-1

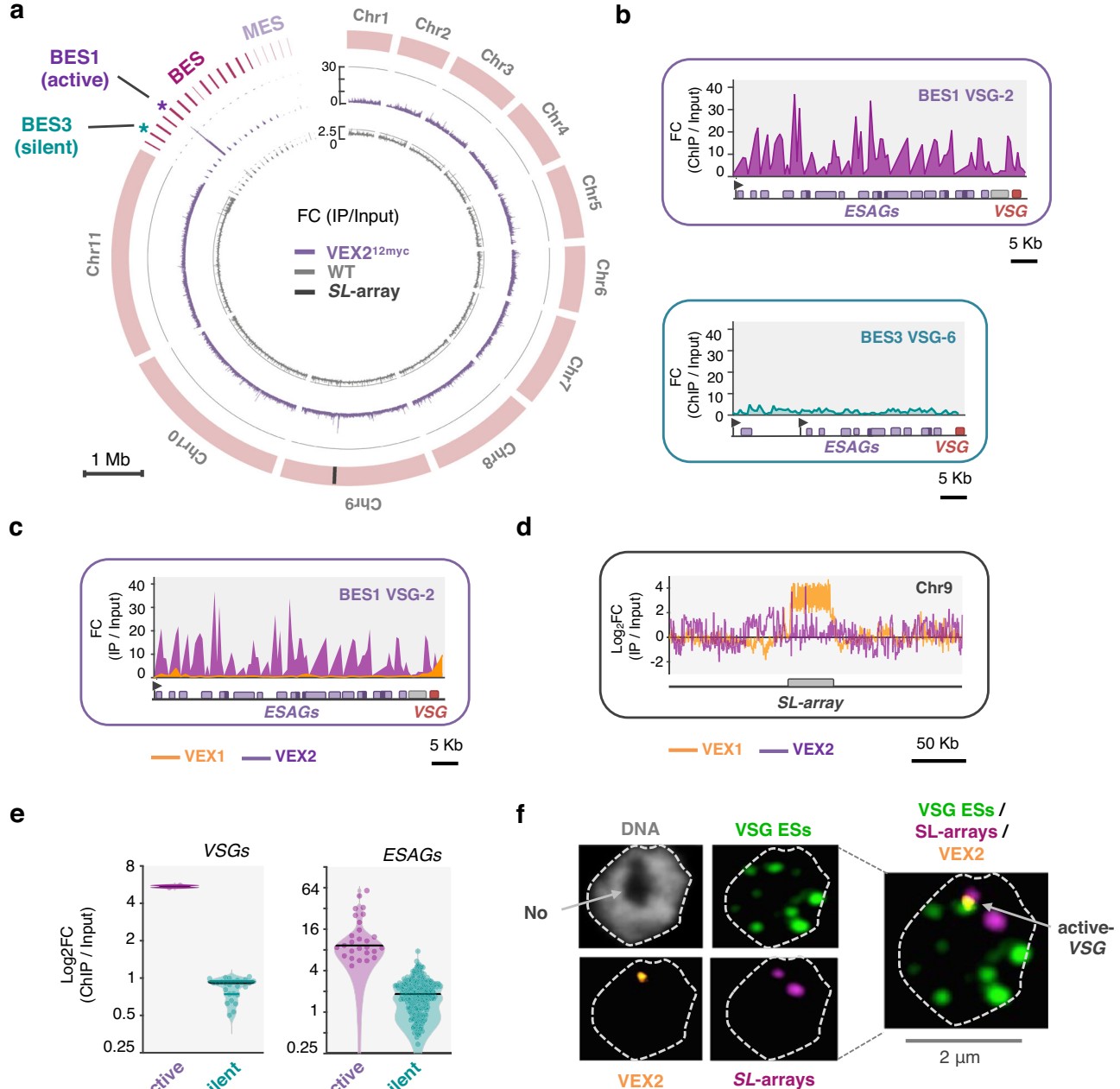

**Fig. 1 | VEX2 is allele-selective. a–e** VEX2$^{myc}$ chromatin immunoprecipitation followed by deep sequencing (ChIP-Seq). VEX2 enrichment across the genome is expressed as ChIP versus input fold change (FC). **a** The circos plot shows VEX2-enrichment (purple) across the *T. brucei* core genome (11 megabase chromosomes), and bloodstream and metacyclic *VSG*-ESs. The inner track shows data for a no myc-tag control (wild-type; grey). The 'active' and the 'silent' expression sites depicted in (**b**) are highlighted. The *SL*-array on chromosome 9 is also highlighted, black line. **b** VEX2-enrichment at the active *VSG*-ES and one 'silent' *VSG*-ES. **c**, **d** VEX2 (purple) and VEX1 (orange) enrichment traces at the active *VSG*-ES (**c**) and at the Spliced Leader-arrays[51] (**d**) are overlayed. **e** The violin plots show VEX2-enrichment across *VSG* or *ESAG* CDSs within bloodstream *VSG*-ESs. **a–e** Bin size 0.5 kb; values are averages of two biological replicates. **f** DNA-FISH combined with immuno-fluorescence followed by super-resolution microscopy. Cells where VEX2 was endogenously tagged with 12xmyc at the C-terminus (orange) were co-stained with biotin and digoxigenin-labelled DNA probes targeting the 50-bp repeats (all *VSG*-ESs, green) and *SL*-arrays (magenta). DNA was stained with DAPI (grey). The images were acquired using a Zeiss LSM880 Airyscan, are representative (80 G1 cells were imaged across two independent experiments), and correspond to maximum-intensity 3D projections; 0.1 μm stacks.

interaction (Supplementary Fig. 5c). The VEX/CAF-1 complexes described above were further validated by size exclusion chromatography. Detection of each component in the eluted fractions revealed high molecular mass VEX1/VEX2/CAF-1 complexes of ~1 MDa and a VEX1/CAF-1 complex of ~0.5 MDa (Supplementary Fig. 5e, f).

Since VEX2 sub-nuclear distribution is tightly restricted, we assessed the rate of VEX1/VEX2 turnover, and whether the interactions impact protein abundance (Fig. 3d, e). VEX1$^{myc}$ had a relatively short half-life (~1 h), which increased after VEX2 knockdown, indicating that VEX2 destabilises VEX1 (Fig. 3d). In contrast, $^{myc}$VEX2 had a relatively long half-life (>4 h), which decreased following VEX1 depletion, indicating that VEX1 stabilises VEX2 (Fig. 3e). Both proteins are turned over by the proteosome (Fig. 3f and Supplementary Fig. 5g, h). Thus, we identified oligomeric forms of the VEX1-VEX2 complex. We show that this complex limits its own abundance through reciprocal controls, whereby VEX1 stabilises VEX2 and VEX2 destabilises VEX1.

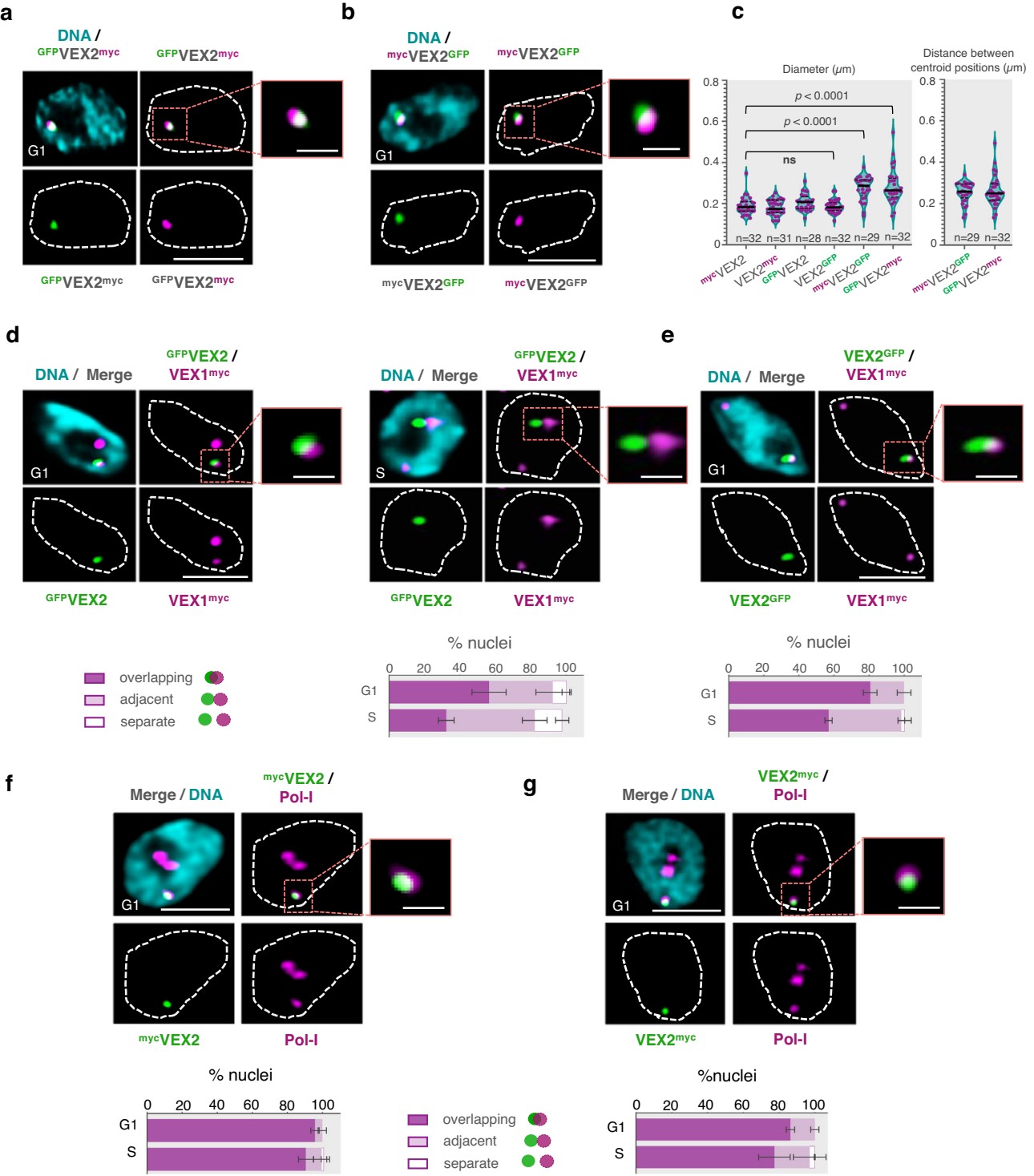

**Fig. 2 | VEX2 bridges loci on distinct chromosomes. a–g** Super-resolution microscopy analysis of $^{GFP}VEX2^{myc}$, $^{myc}VEX2^{GFP}$, $^{GFP}VEX2/VEX1^{myc}$, $VEX2^{GFP}/VEX1^{myc}$, $^{myc}VEX2/Pol-I$ and $VEX2^{myc}/Pol-I$. The images are representative, were acquired using a Zeiss Elyra 7 microscope (Lattice SIM²) and correspond to 3D projections by the brightest intensity of 0.1 μm stacks. DNA was stained with DAPI (cyan); scale bars: 2 μm. **c** The violin plot on the left-hand side indicates the diameter of VEX2 signals when VEX2 is tagged at the N- and/or C-terminus. The black bar corresponds to the median; all datapoints are shown (purple circles). Two-tailed unpaired Student's *t*-test; ns, non-significant. The violin plot on the right-hand side indicates the distance between the centroid positions of VEX2 signals. The data are representative of two independent experiments. **d–g** The graphs represent % of G1 or S phase nuclei (>80 nuclei) where VEX2 N- or C-terminal tag signals overlap, are adjacent or are separate to VEX1 signals (for more details on these assignments, see section 'Microscopy and image analysis'); averages of two biological replicates and representative of two independent experiments; error bars indicate standard deviation.

## The N-terminal fragment of VEX1 interacts with VEX2 to form the bridge

Having established that VEX1 binds chromatin at the splicing locus on chromosome 9 and that VEX2 binds chromatin at a single telomeric *VSG*-ES, we next further elucidated VEX-complex interactions at the interface between the two chromosomes. VEX1 contains regions predicted to promote aggregation (123–131 and 249–255 aa, see Methods) and a SWIM-type zinc-finger (782–812 aa)

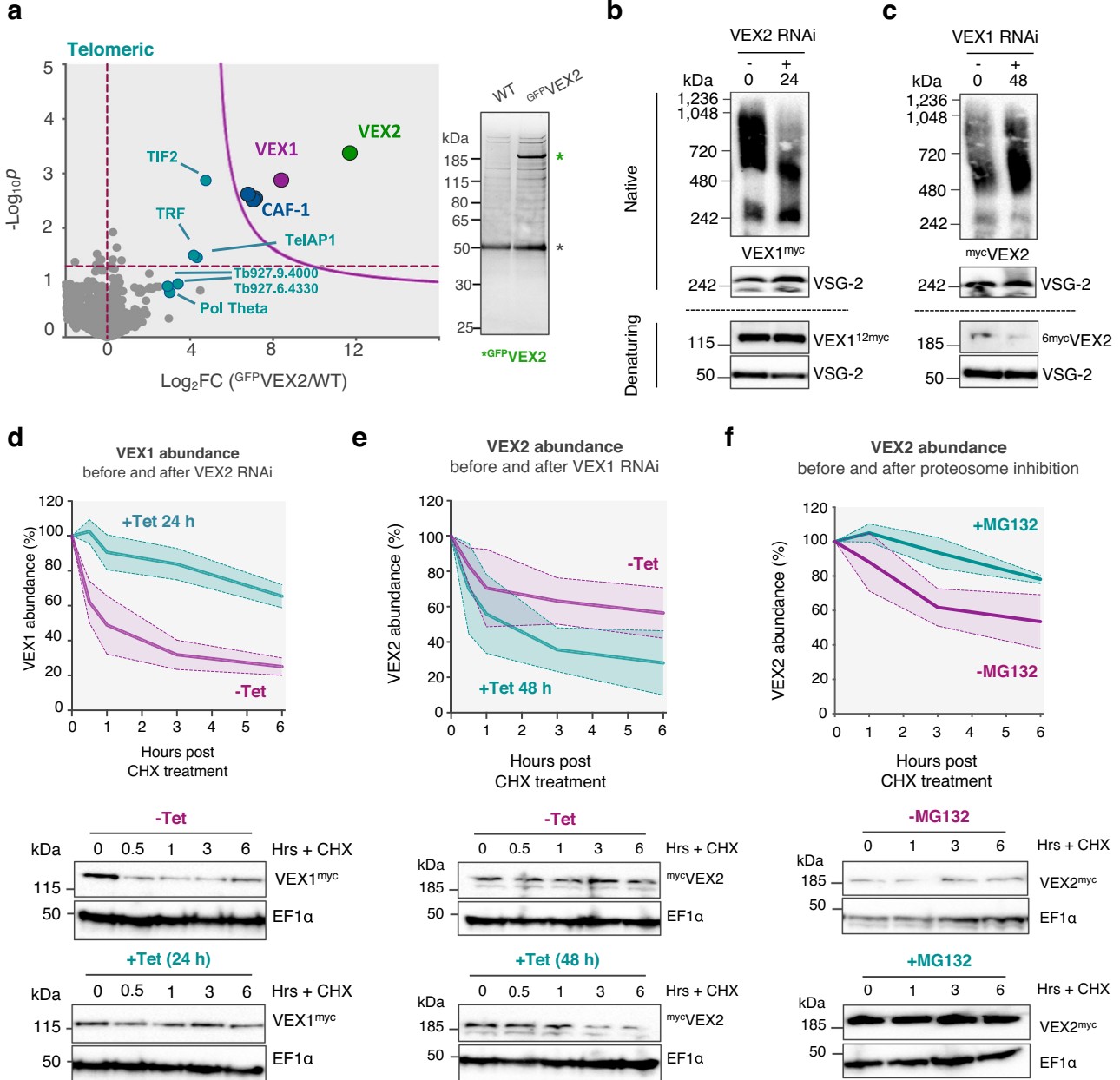

**Fig. 3 | VEX1 and VEX2 form a multimeric and self-limiting complex. a** [GFP]VEX2 affinity purification using a cryo-milling approach followed by LC-MS/MS analysis. The volcano plot depicts log$_2$ fold change (log$_2$FC) versus -log$_{10}p$ (statistical significance) of the LFQ intensities of the tagged cell line and an untagged wild-type control. $p$ values were determined using a two-sided two-sample $t$-test (Perseus v1.5.2.6). The cutoff curve was defined by a permutation-based false discovery rate of 5% and minimum fold change S$_0$ at 2.0. The plots correspond to three independent replicates, using sodium citrate/Tween-20 as extraction buffer. The silver-stained gel on the right-hand side depicts the enrichment of VEX2 in the tagged cell line (green star). The black star indicates α-GFP nanobodies. **b, c** VEX1[myc] and [myc]VEX2 complexes were analysed following VEX2 (**b**) or VEX1 knockdown (**c**).

Native PAGE (upper panels) and denaturing SDS-PAGE (lower panels) blots are shown. VSG-2 was used as control. The blots are representative of at least two independent experiments. **d–f** VEX1[myc] or [myc]VEX2 turnover were monitored following cycloheximide (CHX) treatment (100 μg/mL for 0 to 6 h, at 37 °C) and VEX2 knockdown (24 h tet-induction; **d**), VEX1 knockdown (48 h tet-induction; **e**) or proteosome inhibition by MG132 (5 μM for 1 h, at 37 °C; **f**). Representative protein-blots are shown (**d–f**). EF1α was used as a loading control on the same blots used for VEX1 or VEX2 analysis; -Tet/+Tet or -MG132/+MG132 samples were run in the same gel for each experiment. The graphs show averages of three (**d, e**) or two (**f**) independent experiments; the solid line shows the average and the dashed lines show the upper and lower limits of the standard deviation.

(Fig. 4a) and VEX1 overexpression is known to disrupt VEX2 function[19]. Strains expressing an N-terminal fragment (1–289 aa) or a C-terminal fragment (290–917 aa) were compared to strains overexpressing full-length VEX1, each fused to a 6xmyc tag (Fig. 4 and Supplementary Figs. 6 and 7). Full-length overexpressed VEX1 was distributed throughout the nucleus, although regions that displayed a more intense signal were observed, consistent with accumulation at *SL*-arrays (Fig. 4b). The N-terminal fragment accumulated outside the

nucleus (Fig. 4c, lower panels), suggesting that a nuclear localisation signal (NLS) resides within the C-terminal fragment. A single nuclear focus was observed in cells with the N-terminal fragment expressed at a lower level, however (Fig. 4c, middle panels), and we suggest that this reflects nuclear import and retention through association with native VEX2. The VEX1 C-terminal fragment was distributed throughout the nucleus, similar to the full-length protein (Fig. 4d). These results suggest that a VEX2-interacting domain resides within the N-terminal

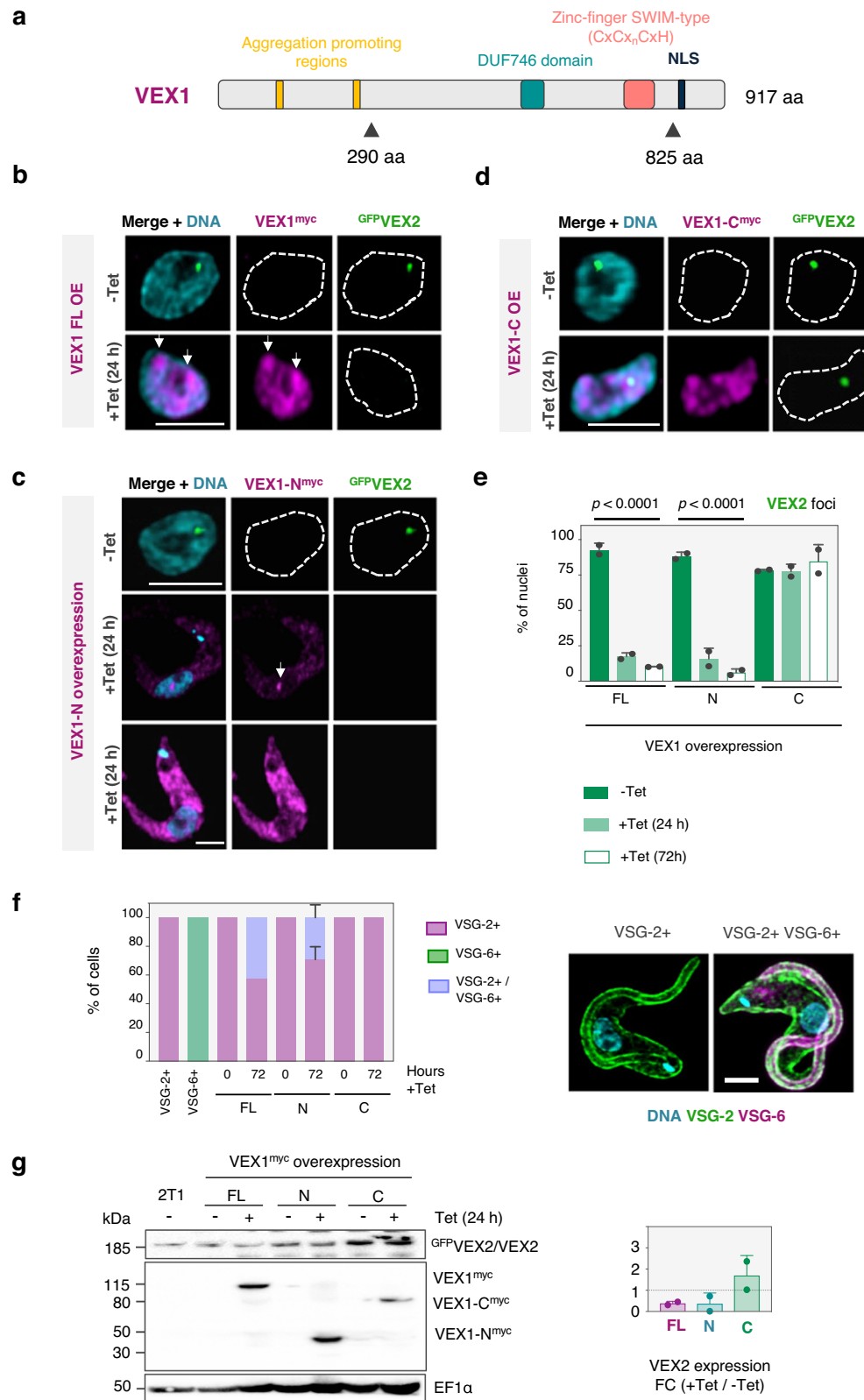

**Fig. 4 | Interactivity across the VEX1-2 bridge. a** Overview of predicted domains and motifs in VEX1. NLS, nuclear localisation signal. FuzDrop was used to predict aggregation-promoting regions. The triangles represent truncation sites.
**b–d** Representative fluorescence microscopy images of GFPVEX2 (green) and over-expressed full-length VEX1 (**b**), N- (**c**) or C-terminal (**d**) regions (fused with 6xmyc, magenta). **e** Impact of overexpression on the % of nuclei with a GFPVEX2 focus. Two-tailed paired Student's *t*-test. **f** Impact of overexpression on monogenic VSG-2

expression. **g** Protein-blot analysis of VEX2 and VEX1 (region) expression. Loading control was EF1α. The graph shows VEX2 abundance versus uninduced controls.
**e–g** Averages of two biological replicates and representative of two independent experiments; error bars correspond to standard deviation. The images in (**b–d**) were acquired using a Zeiss L980 Airyscan 2 (+Airyscan joint deconvolution) and are representative (>100 nuclei quantified) and correspond to maximum-intensity 3D projections of 0.1 μm stacks; DNA was stained with DAPI (cyan); scale bars: 2 μm.

fragment of VEX1 while an NLS resides within the C-terminal fragment. Further analysis showed that the NLS does indeed reside within the C-terminal fragment, between 825 and 917 aa (Supplementary Fig. 6a–c).

VEX2 foci were disrupted by overexpression of either full-length VEX1 (Fig. 4b) or the probable VEX2-interacting N-terminal fragment (Fig. 4c), but not by the C-terminal fragment (Fig. 4d), with VEX2 foci detected in <25% of nuclei in the former two cases (Fig. 4e). As expected, we detected growth defects and cells expressing multiple VSGs following disruption of VEX2-foci by overexpression of either full-length VEX1 or the N-terminal fragment (Fig. 4f and Supplementary Fig. 6d–i). Also consistent with these results, quantitative protein blot analysis revealed that overexpression of either full-length VEX1 or the N-terminal fragment reduced VEX2 abundance (Fig. 4g). Thus, overexpression of the probable VEX2-interacting, VEX1 N-terminal fragment recapitulates the phenotypes associated with overexpression of full-length VEX1.

Our results above suggested that the N-terminal fragment of VEX1 interacts with VEX2. To investigate this hypothesis further, we generated an endogenously tagged copy of the VEX1 N-terminal fragment and observed, by super-resolution microscopy, a single discreet focus that co-localised with VEX2 in over 95% of nuclei (Fig. 5a); a second *SL*-array associated focus was not observed in this case. To further examine this interaction, we performed co-immunoprecipitation using strains expressing full-length VEX1 or its N-terminal fragment tagged at the native locus with 12xmyc, and VEX2 tagged at the native locus with GFP (or not, in a corresponding negative control). For the VEX1 C-terminal fragment, which was undetectable when endogenously tagged, we used a strain overexpressing this fragment, fused with 6xmyc, in a tetracycline-inducible-manner (Supplementary Fig. 6i). Affinity purification of full-length VEX1 or the N-terminal fragment, but not the C-terminal fragment, co-immunoprecipitated VEX2 (Fig. 5b), confirming that the N-terminal fragment of VEX1 mediates the interaction with VEX2.

When overexpressing full-length VEX1, or the N- or C-terminal fragments, we noted a striking difference in protein abundance, with the N-fragment displaying relatively high abundance and the C-fragment displaying relatively low abundance (Supplementary Fig. 6g–i). Since VEX1 displays relatively high turnover (Fig. 3d), we wondered whether this property was mediated by sequences residing in the C-terminal fragment. Indeed, an assessment of relative turnover (Supplementary Fig. 7) revealed that the N-terminal fragment was more stable than full-length VEX1, which was in turn more stable than the C-terminal fragment (Supplementary Fig. 7d, left-hand panel). The N-terminal fragment tagged at the native locus was also more stable than full-length VEX1 tagged at the native locus (Supplementary Fig. 7d, right-hand panel). Thus, a VEX2-interacting and probable bridging domain resides within the N-terminal region of VEX1. The C-terminal region of VEX1 contains an NLS, and the C-terminal fragment of VEX1 analysed here also displays increased turnover. Phenotypes associated with the N- or C-terminal fragments of VEX1 and with VEX2 are summarised in Fig. 5c.

## VEX2 maintains multi-allelic exclusion

VEX2 depletion resulted in *VSG* derepression and, using bulk RNA-Seq, we previously detected transcripts from all *VSG*-ESs[19], but whether several *VSG*-ESs can be simultaneously active in the same cell remained unknown. To explore the capacity of *T. brucei* cells to express multiple *VSG* alleles, and of VEX2 to exclude multiple *VSG* alleles, we performed scRNA-Seq using: wild-type bloodstream-form *T. brucei* expressing VSG-2; a mixture of VSG-2 and VSG-6 expressing cells; VEX1-depleted cells; or VEX2-depleted cells. We also assessed these cell populations for VSG-2 and VSG-6 expression by immunofluorescence microscopy, which revealed cells expressing both VSGs following either VEX1 or VEX2 knockdown (Fig. 6a–c).

Expression of the full set of 22 expression site *VSGs* was analysed using scRNA-Seq data (Fig. 6d–i). The vast majority (99.6%) of VSG-mapping reads from the mixture of VSG-2 and VSG-6 expressing cells identified *VSG-2* or *VSG-6*, as expected (Fig. 6d, g). Following VEX1 knockdown, we observed the derepression of relatively few *VSGs*, typically either *VSG-8* or *VSG-6* (Fig. 6e, h). In contrast, following VEX2 knockdown, several additional *VSG* transcripts were detected, with *VSG-6*, *VSG-8*, *VSG-15* and *VSG-17* being the top-ranking *VSGs* after *VSG-2* (Fig. 6f, i). A full expression profile for all expression site *VSGs* in each of the four populations analysed, and for two replicates for each population, is shown in Supplementary Fig. 8a–d.

To further assess VSG derepression profiles, cells from replicate experiments following VEX2-knockdown were integrated and visualised using UMAP (Uniform Manifold Approximation and Projection)[29]. Cells expressing up to twelve *VSGs* were detected (Fig. 7a), while most cells (4502; 80%) expressed 3–6 *VSGs*, and only 2.6% of cells (146) appeared to maintain monogenic *VSG-2* expression; *VSG* expression profiles were highly reproducible between biological replicates following either VEX1 or VEX2 knockdown (Supplementary Fig. 8e). The results indicated that individual cells could indeed express many *VSGs* simultaneously, and that VEX2 is responsible for multi-allelic exclusion, moreover with many different *VSG* combinations detected in individual cells following VEX2 knockdown (Fig. 7b). We also noted that VEX2-depleted cells displayed greater derepression of *VSGs* in expression sites containing two Pol-I promoters, while 'metacyclic *VSGs*' (VSGs typically activated in the insect to mammal transmission stage metacyclic cells[30]) in monocistronic expression sites, were only moderately derepressed (Figs. 6i and 7c). Thus, VEX2 is required for multi-allelic exclusion, while *VSG*-ES promoters primarily drive derepression during VEX2 depletion. Derepression appears to follow a predictable hierarchy in these cells, with *VSG-6*, *15*, *17*, *18* and *8* dominating (Fig. 7d); scaled profiles for the full set of expression site *VSGs* following VEX2 knockdown, and visualised using UMAP are shown in Supplementary Fig. 8g. Notably, following VEX2 knockdown, total *VSG* levels are substantially increased in cells expressing multiple *VSGs*, while *VSG-2* expression itself is not substantially altered in these cells (Fig. 7e).

The less penetrant phenotype that emerged following VEX1 knockdown was also visualised in low dimensional space using UMAPs (Fig. 8a, b). These results revealed cells either monogenically expressing *VSG-2* (42%; 1002 cells) or also expressing a second *VSG* (41%; 978 cells), typically either *VSG-8* or *VSG-6* (Fig. 8b); scaled profiles for the full set of expression site *VSGs* following VEX1 knockdown, and visualised using UMAP, are shown in Supplementary Fig. 8h. The striking difference between the VEX1 and VEX2 knockdown profiles, and difference in numbers of derepressed *VSGs*, can be seen in Fig. 8c–e.

## Discussion

Antigenic variation is a powerful and sophisticated virulence mechanism with profound impacts on health and disease, but there remain substantial gaps in our understanding of the mechanisms that underpin gene choice, and singular, switchable antigen expression. This is true for several parasites, including those that cause malaria and sleeping sickness[3,4]. More broadly, mechanisms underpinning monogenic expression remain incompletely understood in mammals, where odour detection relies on olfactory receptor allelic exclusion, while B- and T-cell-specific responses rely on immunoglobulin receptor allelic exclusion[31,32]. Similarly, dosage compensation relies on X-chromosome inactivation in female mammals[2].

We previously identified VSG-exclusion-factors-1 and 2 (VEX1 and VEX2) in *T. brucei*[19,20], which associate with the *SL*-array and the active telomeric *VSG*-ES, respectively[13]. Both factors are critical for maintaining, and likely also establishing, *VSG* allelic exclusion. Following chromatin immunoprecipitation analysis, we find that VEX2

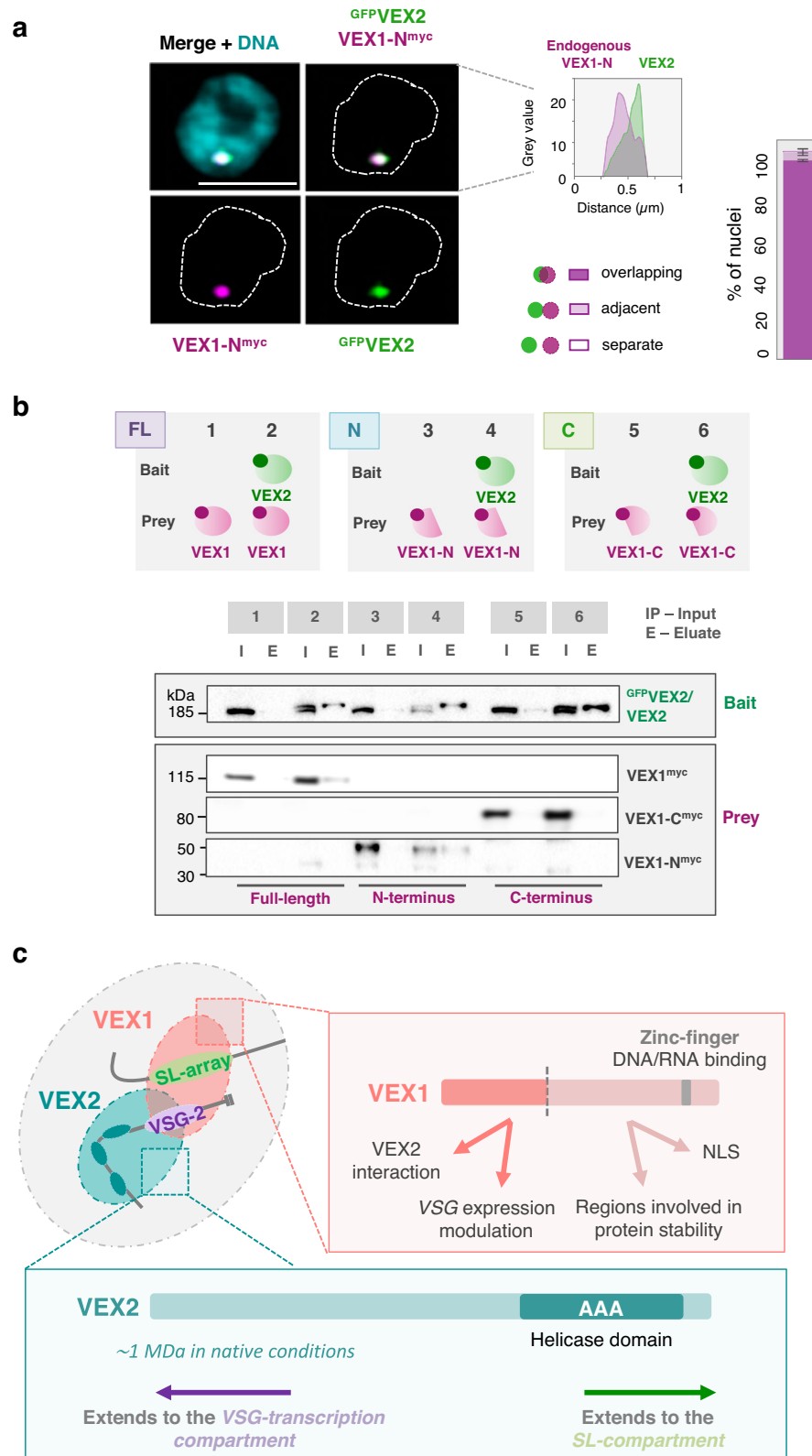

**Fig. 5 | The N-terminal fragment of VEX1 interacts with VEX2. a** Representative images of VEX1-N$^{myc}$ (endogenous level expression) colocalising with $^{GFP}$VEX2. The graph shows the relationship between VEX1 and VEX2 signals in G1 nuclei and shows averages of two biological replicates that are representative of two independent experiments; error bars correspond to standard deviation. The images were acquired using a Zeiss LSM880 Airyscan (>100 nuclei quantified) and correspond to maximum-intensity 3D projections of 0.1 μm stacks; DNA was stained with DAPI (cyan); scale bar: 2 μm. **b** Co-immunoprecipitation of $^{GFP}$VEX2/VEX1$^{myc}$, $^{GFP}$VEX2/ VEX1-N$^{myc}$ and $^{GFP}$VEX2/VEX1-C$^{myc}$ with GFP-Trap magnetic agarose beads (Chromotek) followed by protein-blot analysis; lysis was in HEPES/NaCl buffer and blotting was with α-VEX2 and α-myc. Green, GFP; magenta, myc; I-input; E-elution. All proteins were endogenously tagged, except for the C-terminal region of VEX1, which was overexpressed (24 h induction). The protein blots are representative of two independent experiments. **c** Summary of VEX1 and VEX2-associated features and phenotypes. NLS nuclear localisation signal.

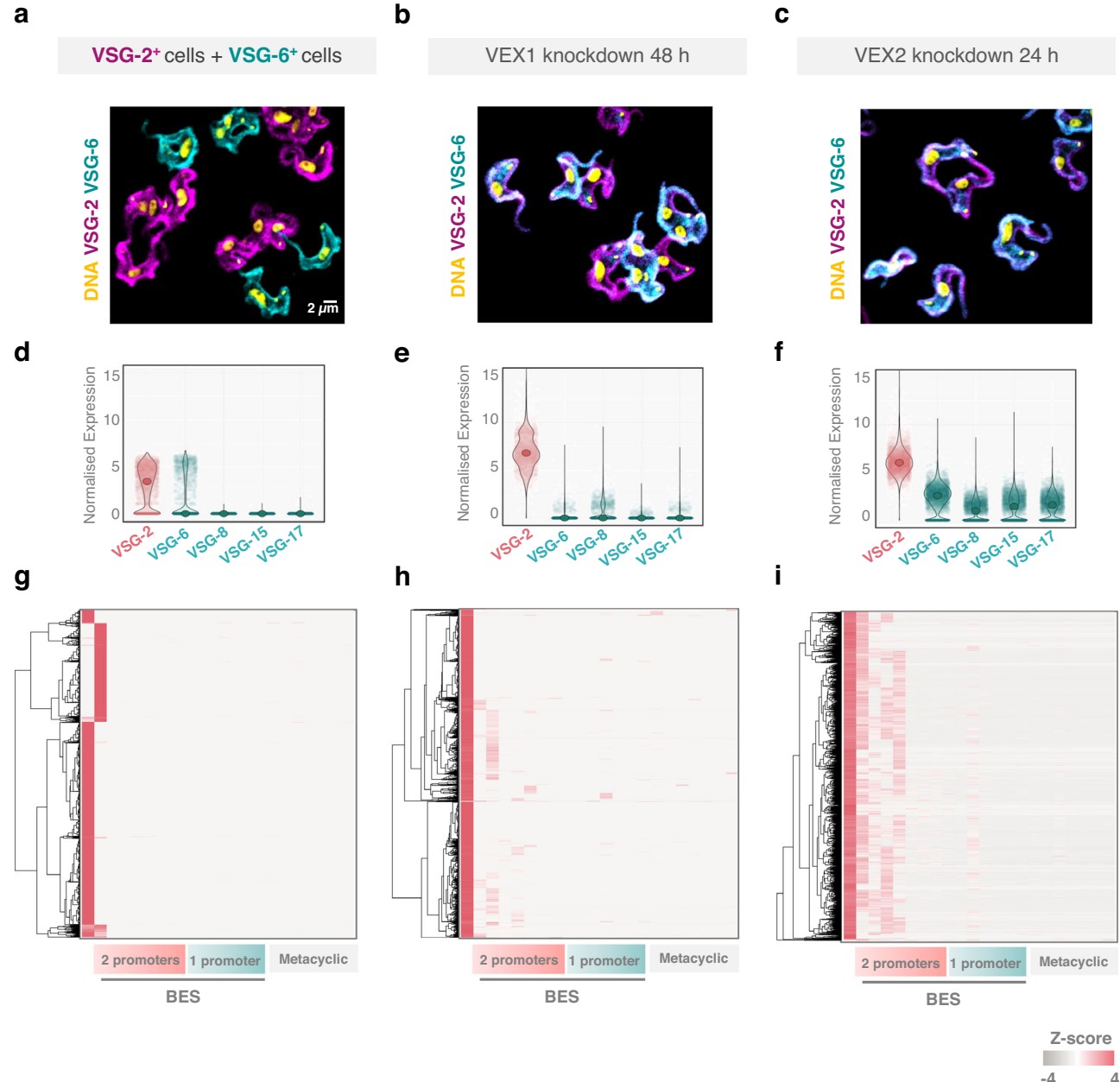

**Fig. 6 | VEX2-depleted cells primarily derepress *VSG*-ESs with two Pol-I promoters.** The images and plots depict a mixture of VSG-2⁺ and VSG-6⁺ cells (**a**, **d**, **g**), VEX1 knockdown (48 h post-induction, **b**, **e**, **h**) or VEX2 knockdown (24 h post-induction; **c**, **f**, **i**). **a**–**c** The immunofluorescence images correspond to maximum-intensity 3D projections generated using a Leica LSM SP8 confocal microscope. Cells were stained with anti-VSG-2 (magenta), anti-VSG-6 (cyan) and DAPI (yellow; DNA). **d**–**f** The violin plots show the expression of the top five derepressed *VSG* genes per cell, for one of two replicates; all datapoints are shown. **g**–**i** The heatmaps show relative expression of *VSGs* from bloodstream expression sites (BES) and metacyclic *VSGs*. Clustering to rows; clustering method: complete linkage; distance measurement: Euclidean. The scale shows the log normalised *Z*-score.

exclusively associates with the active *VSG*-ES, demonstrating that the VEX-complex is allele-selective and bridges genomic loci on two distinct chromosomes. VEX2 is a large protein (224 kDa) that, under native conditions, is present as heterodisperse oligomeric forms of between ~500 and 1200 kDa, suggesting dimeric to hexameric complexes. VEX2 termini can be separately visualised using super-resolution microscopy, with the N-terminus extending towards the active *VSG* transcription compartment, and the C-terminus extending towards an *SL*-splicing compartment.

VEX1 interacts with both VEX2 and Chromatin Assembly Factor 1 (CAF-1)[19]. Affinity purification suggests that these are the sole stable interacting partners; but transient/weaker interactions with telomeric proteins and transcription elongation factors are also detected, indicating proximity to telomeric and gene expression machinery. We have suggested that inheritance of monogenic *VSG* expression requires VEX1 to reload following DNA replication, and that VEX1 reloading is CAF-1 dependent[19]. Consistent with this view, we show that a pool of VEX1 associates with CAF-1. In contrast, and consistent with CAF-1 independent sequestration of VEX2[19], VEX2 does not appear to directly interact with CAF-1. Furthermore, the VEX1 and VEX2 association is dynamic and cell cycle dependent, as demonstrated by both imaging and biochemical analysis.

Our results demonstrate that the VEX2-interacting region of VEX1 lies within the N-terminal fragment (1–289 aa), and that while VEX1

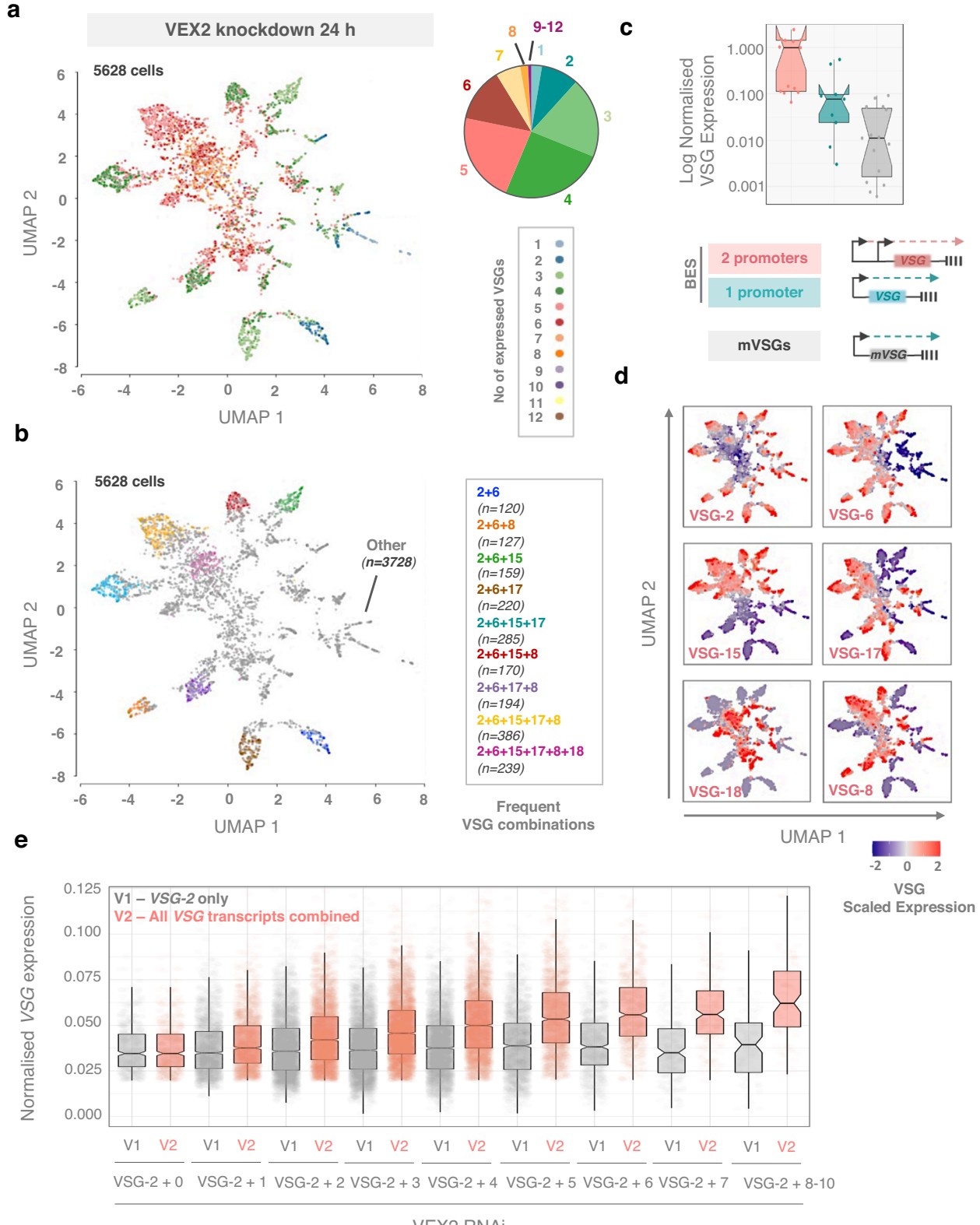

stabilises VEX2, VEX2 destabilises VEX1; turnover of both proteins is proteasome-dependent. Notably, since native VEX1 stabilises VEX2, while overexpressed VEX1 destabilises VEX2, we suggest that VEX1 selectively stabilises VEX2 in the context of an active *VSG* expression site. Thus, self-limiting control of the VEX-complex could facilitate allele-selective chromosomal bridging and limit the pool of VEX2 available to accumulate at other *VSG* expression sites. Further

analysis following SLAP1 knockdown suggested that *SL*-RNA and *trans*-splicing are required to maintain the VEX-complex bridge.

VEX2 depletion results in derepression of otherwise silent *VSG*s[19]. Using both bulk RNA-Seq and proteomic analysis of cell-surface VSGs, we detected transcripts and proteins from all *VSG*-ESs, confirming that the transcribed *VSG* genes yielded proteins that were delivered to the cell surface[19]. Whether *VSG* genes were being derepressed

**Fig. 7 | scRNA-Seq reveals multi-allelic exclusion by VEX2. a, b** Low dimensional (UMAP) plots of single cells following VEX2 knockdown (after filtering). Each point is the VSGome (sum of the *VSG* transcripts in the transcriptome) of one cell positioned according to similarity with neighbouring VSGomes, coloured by number of *VSGs* expressed per cell (**a**) or expression of specific *VSG* combinations (**b**). The pie chart in (**a**) shows the % of cells expressing 1 to >9 VSG transcripts per individual cell following VEX2 depletion. **c** The box plot shows the expression of *VSG* genes from bloodstream expression sites (BES) with one or two Pol-I promoters, or from metacyclic expression sites. Each datapoint corresponds to the average expression per cell for each *VSG* (datapoints for each replicate were individually plotted).
**d** UMAP plots of VEX2-depleted parasite VSGomes coloured by transcript counts

for VSG-2 ('active') and five derepressed VSGs (VSG-6, VSG-15, VSG-17, VSG-18, VSG-8). Expression levels are individually scaled for each *VSG* transcript, based on the corresponding minimum and maximum values. **e** The box plot shows normalised *VSG-2* (grey) and total *VSG* (salmon) expression in cells expressing *VSG-2* only or cells expressing *VSG-2* + x number of other *VSGs* in VEX2-depleted cells. All datapoints are shown and represent individual cells from two biological replicates. In (**c**) and (**e**), the box indicates the interquartile range (IQR), the whiskers show the range of values that are within 1.5×IQR and a horizontal line indicates the median. The notches represent for each median the 95% confidence interval (approximated by $1.58 \times \text{IQR}/\text{sqrt}(n)$).

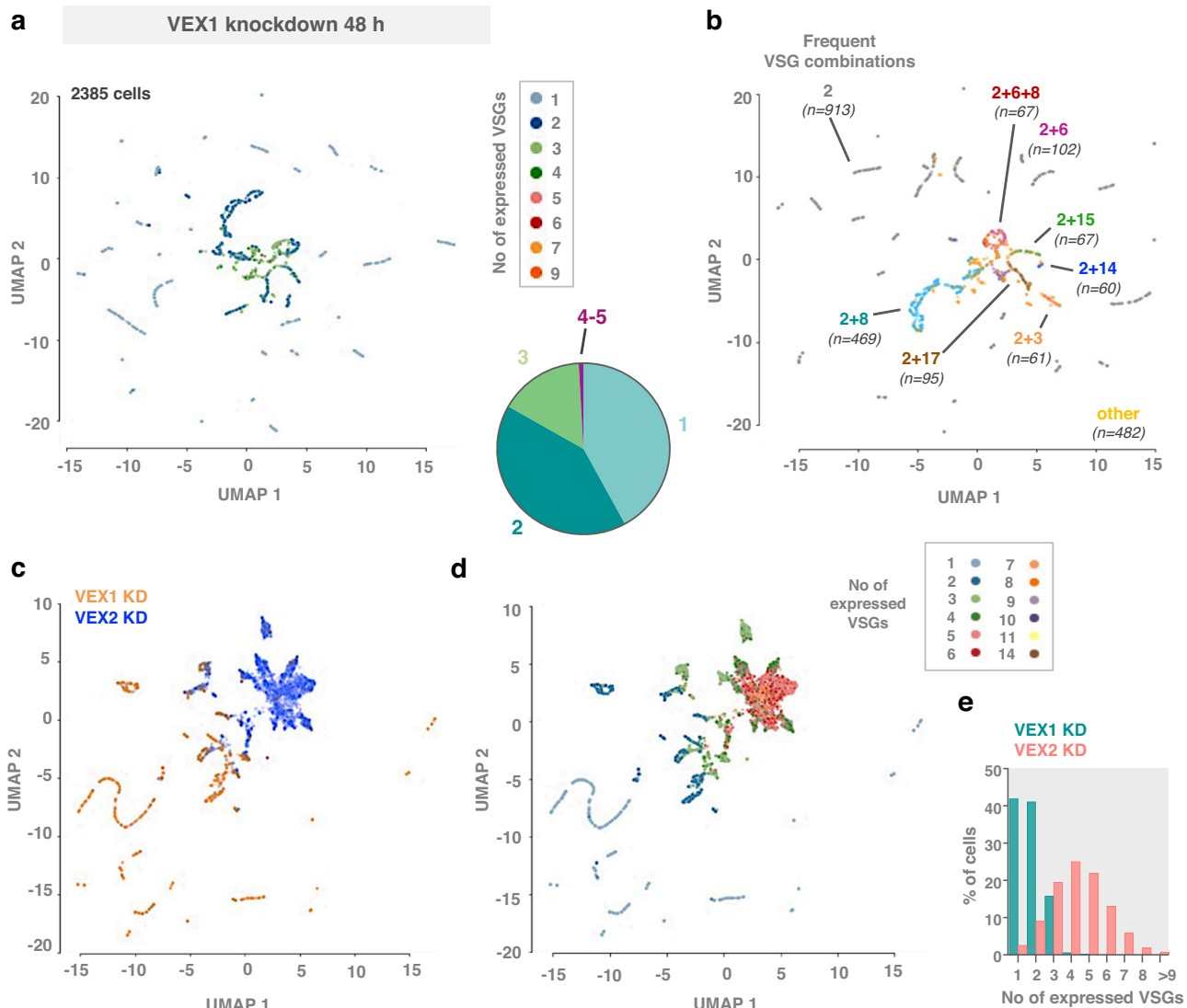

**Fig. 8 | Distinct patterns of VSG derepression in VEX1 or VEX2-depleted cells.**
**a, b** Low dimensional (UMAP) plots of single cells following VEX1 knockdown (after filtering). Each point is the VSGome (sum of the *VSG* transcripts in the transcriptome) of one cell positioned according to similarity with neighbouring VSGomes, coloured by number of *VSGs* expressed per cell (**a**) or expression of specific *VSG* combinations (**b**). The pie chart in (**a**) shows the % of cells expressing

1–5 VSG transcripts per cell following VEX1 depletion. **c, d** Low dimensional (UMAP) plots of each cell following VEX1 or VEX2 knockdown (after filtering). **c** Cells are coloured based on their origin: VEX1 knockdown dataset (orange) or VEX2 knockdown dataset (blue). **d** Cells are coloured based on the number of *VSGs* expressed per cell. **e** The histogram represents the % of cells expressing 1 to >9 VSG transcripts per cell following VEX1 (cyan) or VEX2 (salmon) knockdown.

simultaneously or individually in single cells remained unclear, however, due to the limitations of population-based analyses[33]. We addressed this question using single-cell transcriptomics. Following VEX2-depletion, up to twelve distinct *VSGs* could be detected per cell. However, the most common phenotype was less promiscuous with simultaneous expression of three to six *VSGs* per cell.

*VSG* derepression was more pronounced for those expression sites with two Pol-I promoters, suggesting that *VSG* expression site promoters drive the competition for activation, likely supported by the transcriptional activator ESB1, which recruits Pol-I to the ESB[34]. Notably, both Pol-I reservoirs (the ESB and the nucleolus) dispersed following VEX2 depletion[19], consistent with *VSG* expression site

promoters competing for recruitment of Pol-I. Moreover, amongst *VSG*-ESs with two Pol-I promoters, there is a hierarchy of depression (*VSG-6 > 15 > 17 > 18 > 8*), indicating that other factors contribute, for instance nuclear positioning, or accessibility of the telomeric sites on several distinct chromosomes where these *VSGs* are located; the *VSG-2, 6, 18* and *8* expression sites are on megabase chromosomes, while the *VSG-15* and *17* expression sites are on intermediate chromosomes[6]. Following VEX1 depletion, the derepression phenotype was less pronounced, and although we detected all expression site-associated *VSGs* in the population[20], at the single cell level, most cells derepressed only one *VSG*, typically *VSG-8* or *VSG-6*. Our observations demonstrate that VEX2 is required for the simultaneous exclusion of most, if not all, *VSG* expression sites. Since simultaneous expression of multiple VSGs is associated with a fitness cost and is eventually lethal (~72 h post VEX2 RNAI-induction[19]), we suggest that transcriptional and/or RNA processing machineries become limiting under these conditions.

The C-terminal region of VEX2 contains a putative RNA helicase domain, related to UPF1 (superfamily 1)[35]. Several observations suggest a conserved role for this family of helicases in regulating long non-coding RNA (lncRNA)-chromatin associations, and gene expression choices. For instance, murine UPF1 is required for formation of the RNA domains in the X-inactive specific transcript (*Xist*), a major effector of X-chromosome inactivation[36]. Furthermore, mammalian Rent1 is enriched at telomeres and negatively controls the association between telomeric repeat-containing RNA (TERRA) transcripts and chromatin at these sites, which is linked to heterochromatin assembly[37]. Since the UPF1 family contains both RNA and RNA:DNA helicases[35], it will be important to assess alternative VEX2 substrates. It is also noteworthy that members of this helicase family have been implicated in R-loop resolution, telomere stability and maintenance, regulation of transcription attenuation and termination, splicing regulation, and moderation of transcription/replication encounters[37–40]. It is also worth noting here that other proteins have been implicated in chromosome looping, bridging and nuclear organisation, such as CTCF in metazoa[41,42], but these proteins are not thought to be allele-selective.

In conclusion, we show that VEX2 is allele-selective, and that the VEX-complex controls multi-allelic exclusion by physically bridging two chromosomes. VEX2 liaises the active *VSG* gene on one chromosome, and through interaction with VEX1, provides a bridge to a *trans*-splicing locus on a distinct chromosome. Current results support the view that the bridge is maintained by both transcription and splicing. By examining *VSG* expression following VEX2 depletion at the single cell level, we demonstrated that VEX2 is required for the simultaneous exclusion of multiple *VSG* expression sites, and also that *VSG* expression site promoters drive the competition for expression site activation. Taken together, our results reveal a remarkable inter-chromosomal bridge that supports allelic exclusion, substantially advancing our understanding of the molecular mechanisms underpinning single *VSG* gene choice.

## Methods

A detailed list of reagents, cell lines and software used in this study is provided in Supplementary Data 1, sheet 6.

### *T. brucei* growth and manipulation

Bloodstream-form *T. brucei*, Lister 427 (L427), 2T1 cells[43] and 2T1/T7/Cas9 cells[44] were grown in HMI-11 medium and genetically manipulated using electroporation[45]; cytomix or human T cell nucleofector solution (Lonza) were used for all transfections. Puromycin, phleomycin, hygromycin, neomycin and blasticidin were used at 2, 2, 2.5, 2 and 10 μg mL$^{-1}$ for selection of recombinant clones; and at 1, 1, 1, 1 and 2 μg mL$^{-1}$ for maintaining those clones, respectively. Cumulative growth curves were generated from cultures seeded at 10$^5$ cells mL$^{-1}$,

counted on a haemocytometer and diluted back to 10$^5$ cells mL$^{-1}$ as necessary. Tetracycline was applied at 1 μg mL$^{-1}$ for RNAi or Cas9 induction. Cycloheximide (translation inhibitor, stock 100 mg/mL in DMSO) was applied at 100 μg/mL for 0–6 h and MG132 (proteosome inhibitor) was applied at 5 μM for 1 h, both at 37 °C[46].

### Plasmids and constructs

For the RNAi experiments, previously generated plasmids were used to target VEX1 (Tb927.11.16920) or VEX2 (Tb927.11.13380)[19,20]. For *SLAP1* RNAi, primers were selected from ORF sequences using RNAit[47]. A specific RNAi target fragment for *SLAP1* (Tb927.11.9950, 624 bp) was amplified and cloned in pRPa$^{iSL}$[48] (primers P25 + P26; Supplementary Data 1, sheet 5). For epitope-tagging at the native locus[48], some previously generated plasmids were used: pNAT$^{TAGx}$ to add an N-terminal 6× c-myc or GFP-tag to VEX2 and pNAT$^{xTAG}$ to add a C-terminal 12× c-myc or GFP-tag to VEX1, CAF-1b and VEX2, respectively[19,20]. In addition, two more constructs were generated for the present study. pNAT$^{xTAG}$ to add a C-terminal 12× c-myc to VEX2; a fragment of 918 bp was amplified (primers – P1 + P2, Supplementary Data 1, sheet 5) and cloned in pNAT$^{xTAG}$, which was then linearised with HpaI prior to transfection. To add N-terminal 6x c-myc to SLAP1; a fragment of 900 bp was amplified (primers P3 + P4, Supplementary Data 1, sheet 5) and cloned in $^{TAGx}$pNAT, which was then linearised with NcoI prior to transfection. In order to knockout one VEX2 allele, a replacement cassette was generated by fusion PCR. VEX2 5' and 3'-UTRs (primers: P9-P12) were amplified from *T. brucei* L427 genomic DNA. The amplified *NEO* sequence had 21 and 27 nucleotides of homology to the end and the beginning of VEX2 5' and 3'-UTR, respectively (primers: P13 + P14). The three fragments generated by individual PCRs were purified, mixed in equimolar concentrations, and fused using the P9 + P12 primers: (primers – Supplementary Data 1, sheet 5). The PCR reaction was performed in standard conditions using Phusion polymerase except that for the first 10 cycles a temperature gradient was applied: 0.5 °C decrease in the annealing temperature per cycle (60–55 °C); the annealing temperature was maintained at 55 °C for the next 30 cycles. The resulting PCR product was purified, cloned in pGEM-T-easy and sequenced. The resulting replacement cassette was excised with NotI and Acc65I prior to transfection. Successful replacement of one *VEX2* allele was confirmed by PCR, using a primer upstream of the 5'-UTR sequence in the replacement cassette and another in the *NEO* gene (primers: P15 + P16; Supplementary Data 1, sheet 5. Further confirmation was achieved by Southern blotting (see below). In the absence of major identifiable domains for VEX1, and since it forms discreet sub-nuclear foci and contains a few predicted disorganised regions, we used FuzDrop[49] to predict potential aggregation regions. Based on predicted aggregation sites (123–131 and 249–255 aa) and predicted protein secondary structures, we decided to 'split' the protein in two fragments. We then generated cell lines expressing the 'N-terminal fragment' (1–289 aa) containing the predicted aggregation regions and the 'C-terminal fragment' (290–917 aa) containing a DUF domain and SWIM-type zinc-finger. N-terminal or C-terminal truncated versions of VEX1 were cloned in the over-expression plasmid pRPa (x6myc tag at the C-terminus) or the endogenous tagging plasmid pNAT (x12myc at the C-terminus), using primers P5-P8 (primers – Supplementary Data 1, sheet 5).

### CRISPR/Cas9 editing

For CRISPR/Cas9 editing, the sgRNAs were cloned into pT7$^{sgRNA}$[44]. The plasmid was sequenced for confirmation and then digested with NotI prior to transfection into a 2T1/T7/Cas9 parental cell line. Where applicable, templates were provided transiently using 10–15 μg of a PCR product or 40 μg of a single-stranded oligonucleotide and Cas9 was induced with tetracycline immediately after transfection. The sequences of the oligonucleotides used in these experiments can be found in Supplementary Data 1, sheet 5 (P17-P20). Both alleles of VEX2

were tagged without selectable markers and without disrupting untranslated sequences[24].

## Single-cell RNA-Seq

Four cell lines were used for scRNA-Seq experiments: wild-type (VSG-2+ cells and VSG-6+ cells) and Tet-inducible VEX1 or VEX2 RNAi (two independent clones); densities were kept below ~1 × 10^6 cells/mL. Eight samples were prepared: VSG-2+ cells (×2), VSG-2+ + VSG-6+ cells (50:50 ratio, ×2), VEX1 RNAi 48 h (two independent clones) and VEX2 RNAi 24 h (two independent clones). Samples were prepared as described[50]. Briefly, cells were centrifuged, and the pelleted cells were washed twice with 1 mL ice-cold 1X PBS supplemented with 1% D-glucose (PSG) and 0.04% bovine serum albumin (BSA). Cells were then resuspended in ~500 μL PSG + 0.04% BSA, filtered with 40 μm Flowmi™ Tip Strainer (Merck) and adjusted to 1000 cells/μL. In all steps, cells were centrifuged at $400 \times g$ for 10 min and handled gently using wide-bore tips (Starlab). In total, 15,000 cells (15 μL) from the mixed sample were loaded into the Chromium Controller (10× Genomics) to capture individual cells with unique barcoded beads. Libraries were prepared using the Chromium Next GEM Single Cell 3′ GEM, Library & Gel Bead Kit v3.1, Chromium Next GEM Chip G Single Cell Kit and Single Index Kit T Set A (10× Genomics). Firstly, two samples were analysed at the Tayside Centre for Genomic Analysis by Illumina MiSeq for a preliminary assessment: 12.5 million reads per sample (~1250 reads/cell); read 1 at 28 bp followed by 272 bp for read 2. Deeper sequencing for all samples was then carried out using a DNB-Seq 2000 platform (BGI, Hong Kong), which generated approximately 350 million reads per sample (~35,000 reads/cell); paired-end reads (PE28 + 150 + 8). The 50:50 mixture of VSG-2+ and VSG-6+ cells was used to assess the proportion of doublets in the experiment, which was approximately 12% in the first replica, and approximately 7.6% in the second replica, similar to 10× Genomics' prediction in mammalian cells, and 8% reported following mixing *T. brucei* and *Leishmania mexicana*[50]. The reference genome was compiled with Cell Ranger v3.0.2, to combine the *T. brucei* TREU927 nuclear reference genome assembled by Briggs et al., to extend the 3′-UTR annotations[50,51], *T. brucei* L427 maxi circle kDNA sequence (GenBank: M94286.1[52]) and the bloodstream[8] and metacyclic *VSG*-ESs[22,23] from the L427 strain. Reads were mapped, and unique reads aligned to each annotated gene were counted and assigned to a cell barcode with the Cell Ranger count function. Cell Ranger 6.0.1 (http://software.10xgenomics.com/single-cell/overview/welcome) was used with all default settings. The STAR aligner in Cell Ranger is an intron aware aligner[53]. However, the *T. brucei* genome is almost devoid of introns[54], with a high percentage of repetitive DNA sequences[55]. Using STAR with default settings would create mapping artefacts, with reads mapped across genes sharing high similarities; this was overcome (with advice from the 10× support service) by adding the line '--alignIntronMax=3 --alignIntronMin=2' to the parameter file 'parameters.toml' after the line 'min_fraction_whitelist_match = 0.1'. When running Cell Ranger, we used the EmptyDrops fuction in the DropletUtils R package (1.14.1)[56] to specify a custom lower bound on the total UMI count, and selected the number of real cells with an FDR less than 1%. This number was used as the --force-count option value to control for the presence of ambient RNA contamination, resulting from cell disruption during sample preparation. Per cell, 658 and 588 median genes were detected in each replicate experiment of VEX1 RNAi and 819 and 837 median genes were detected in each replicate experiment of VEX2 RNAi (versus Briggs et al.[50]: 1052 and 1445 genes per cell using 10×; Vigneron et al.[57]: 298 genes per cell using 10×; Muller et al.[6]: 1572 genes per cell using SMART-Seq). We then applied SoupX (1.5.2)[58] to remove ambient RNA contamination from the count data. To ensure that we had captured high-quality transcriptomes, we performed the following controls and applied the following filters during the analysis. First, we confirmed whether single-cell behaviour had been achieved by determining the percentage of doublets using the 50:50 mixture of

VSG-2+ and VSG-6+ cells (~10% and similar to ref. 50). Second, we fed the decontaminated counts to the SingleCellExperiment R package (1.16) and used the perCellQCMetrics function in the scuttle R package (1.4) to add per-cell quality control metrics. We then filtered out with the isOutlier function in the scater (1.22) R package: (1) cells with too many transcripts (>3 median-absolute-deviations), (2) cells with few transcripts (<1 median-absolute-deviations), (3) cells with few detected genes (<1 median-absolute-deviations), (4) cells with a low count of VSG transcripts (<45 absolute count), (5) cells with a high level of mitochondrial transcripts (>2 median-absolute-deviations), (6) cells with high non-polyadenylated ribosomal RNA (rRNA) levels (>2 median-absolute-deviations). After filtering we obtained data for 2544 and 3084 cells for the VEX2 RNAi experiments and 1677 and 708 cells for the VEX1 RNAi experiments, equivalent to 7–30% of the input. The UMAP reduction plots were computed using only the VSG count data after normalisation and log$_{10}$ transformed using the scran (1.22.1) R package and visualised with the plotReducedDim and plotDimReduceFeature functions in the celda (1.10) R package.

The analyses were conducted in Jupyter notebook (6.4.6) using R (4.1.0) and IPython (7.29) environments. To analyse *VSG* expression, we employed a custom script developed in a Jupyter notebook[59]. The primary *VSG* counts were adjusted to the total gene counts within a cell. This normalisation accounted for general transcriptional activity and mitigated any discrepancies in sequencing depth across cells. Cells were then grouped based on their concurrent *VSG* expression profiles, such as primary *VSG* with one additional *VSG*, primary *VSG* with two additional *VSGs*, and so on, up to primary *VSG* combined with 13 other *VSGs*. We then compared the normalised expression levels of the primary *VSG* across these groups to discern how concurrent *VSG* expression might impact the expression levels of the main *VSG*. Subsequently, a second analysis was conducted where the same cell categories were assessed for their overall *VSG* gene content, also normalised to the total gene counts within that cell.

## ChIP-Seq

Chromatin immunoprecipitation and cell lysis were carried out as in ref. 60 with the following modifications. Briefly, $2 \times 10^8$ *T. brucei* bloodstream-form cells expressing a C-terminal 12-myc tagged endogenous copy of *VEX2* (2 independent biological replicates) and wild-type cells (untagged: background control) were cross-linked with 1% formaldehyde for 20 min at room temperature (RT). DNA was sonicated using a Bioruptor (Diagenode) with sonication beads (Diagenode) for 10 cycles of 30 s on/30 s off. C-terminal 12-myc VEX2 was immunoprecipitated with α-myc antibody (NEB, clone 9B11) coupled to Dynabeads Protein G (2.8 μm). Antibody coupling to the Dynabeads was carried out according to the manufacturer's recommendations. The beads were then washed with RIPA buffer (50 mM HEPES-KOH, pKa 7.55, 500 mM LiCl, 1 mM EDTA, 1.0% NP-40, 0.7% Na-Deoxycholate) and the eluted DNA was purified by phenol chloroform extraction and ethanol precipitation. For each biological replicate, two ChIP reactions were performed in parallel and the eluted DNA pooled prior to extraction and purification. The DNA concentration in the sample was determined with the Qubit fluorometer, using the Qubit dsDNA HS kit (Thermo). The samples were sequenced using a DNB-Seq platform (BGI, Hong Kong); 50 bp single-end sequencing reads; approximately 20 million reads per sample. Reads were mapped to the core genome of the TREU927[51] and the bloodstream[8] and metacyclic *VSG*-ESs[22,23] from the L427 strain. Reads were also mapped to the 2018 assembly of the L427 strain[6]. Bowtie 2-mapping[61] was with the parameters --very-sensitive --no-discordant --phred33. Alignment files were manipulated with SAMtools[62]. Alignments were inspected with the Artemis genome browser[63]. The resulting BAM files were loaded into Galaxy[64] for downstream analysis. PCR duplicate reads were removed using Picard MarkDuplicates (https://broadinstitute.github.io/picard/) and the reads were filtered with mapQ > 0 for the core genome and

mapQ > 1 for *VSG*-ESs. The ratio of ChIP/Input across the genome was generated using BamCompare (deepTools[65]) with a bin size of 500 bp or 1 kb, generating bedgraph files as outputs. Peak calling was conducted using MACS2[66]—a threshold (minimum *p* and *q* values to define a peak) was defined by comparison with the untagged control. Reads per CDS feature were extracted using featureCount[67], normalised to the total amount of reads per sample, and then used to calculate the ratio of ChIP/Input. Radial plots were generated using Circos[68] and further edited using Adobe Illustrator (https://www.adobe.com), while *VSG*-ES plots, a heatmap and violin plots were generated using GraphPad Prism V9.0 (https://www.graphpad.com).

## Southern blotting

Genomic DNA was extracted using a DNA extraction kit (Qiagen) according to the manufacturer's instructions. Approximately 10 µg of genomic DNA was digested with a 5-fold excess of AgeI and EcoRV or EcoRV and PsiI overnight (O/N) at 37 °C. The samples were then run O/N in a 0.8% agarose gel. The gel was sequentially incubated with 0.25 M HCl, 1.5 M NaCl 0.5 M NaOH and 3 M NaCl 0.5 M Tris.HCl pH 7. DNA was then transferred O/N onto a Nylon membrane (Amersham), using 10× SSC (saline sodium citrate: 300 mM sodium citrate, 1 M NaCl). Nucleic acids fixation was achieved at 65 °C for 5 h. Hybridisation and development were performed using the DIG high prime DNA labelling and detection starter kit II (Roche) following the manufacturer's instructions. The probe was applied at 25 ng/mL. VEX2 5′-UTR (primers: P9 + P10) or 3′-UTR (primers: P11 + P12) were used as probes (primers – Supplementary Data 1, sheet 5).

## Affinity purification of VEX2 interacting proteins

Bloodstream-form *T. brucei* cells ($1.2 \times 10^{10}$), with or without N-terminal GFP-tagged endogenous copies of *VEX2* (from double allele tagging using CRISPR/Cas9[24]) were washed three times in ice-cold PBS with protease inhibitor cocktail (Roche, EDTA free) and subsequently cryo-milled in a planetary ball mill (Retsch) into a fine powdered grindate[69]. Initial screening for the most suitable extraction buffer required 50 mg of frozen grindate per sample suspended in ice-cold lysis buffer (20 mM HEPES, pH 7.4, 1 mM MgCl₂, 10 µM CaCl₂, protease inhibitor cocktail) containing either 100 mM NaCl or 250 mM citrate and either 0.1% Tween-20 or 0.1% Brij58. The samples were then subjected to sonication on ice (10 cycles of 3 s on/10 s off at 40% amplitude). Samples were then spun for 10 min at 20,000 *g* at 4 °C to pellet cellular debris. The supernatant was removed and added to a low protein binding 1.5 mL Eppendorf tube containing 4 µL GFP-Trap magnetic agarose beads (Chromotek) and agitated at 4 °C for 2 h. The samples were then placed on a magnetic rack and washed three times with ice-cold lysis buffer. Samples were eluted with NuPAGE LDS loading buffer containing reducing agent (Invitrogen) at 70 °C for 10 min. Next, the resulting proteins were fractionated by SDS-PAGE and visualised by silver staining. Subsequently, three (NaCl/Tween-20; Citrate/Tween-20) independent pull-downs were performed and subjected to liquid chromatography coupled with tandem mass spectrometry (LC-MS/MS) analysis. For this purpose, the process described above was scaled-up. Six aliquots of 50 mg of frozen grindate were used per condition, and six immunoprecipitation reactions conducted in parallel—the six samples were pooled together in the last wash. The conditions were as above, except for sonication (20 cycles were used instead of 10). After eluting the samples, these were run 2 cm into a NuPAGE 4–12% Bis-Tris gradient gels (Invitrogen) and protein-containing slices were excised.

## Quantitative mass spectrometry

Gel slices containing affinity-enriched samples were first washed to remove detergents and buffer salts using MilliQ water, acetonitrile, 100 mM of ammonium bicarbonate, a 50:50 mixture of acetonitrile and 100 mM of ammonium bicarbonate, and lastly,

acetonitrile (15 min each). The samples were then reduced, alkylated and subjected to overnight (>16 h) trypsin digestion (Pierce). Peptides were then extracted, dried in a SpeedVac (Thermo Scientific), resuspended in 50 µL 1% formic acid/2% acetonitrile, centrifuged and transferred to high-performance liquid chromatography (HPLC) vials. Samples (10 µL) were typically analysed on a LTQ OrbiTrap Velos Pro (Thermo Scientific) coupled to an UltiMate 3000 RSLCnano ultra HPLC system (Thermo Scientific) and EasySpray column (75 µm × 50 cm, PepMap RSLC C18 column, 2 µm, 100 Å, Thermo Scientific). The mass spectrometer was operated in data-dependent mode with a single MS survey scan from 335 to 1800 *m/z* followed by 20 sequential *m/z* dependent MS2 scans. The 20 most intense precursor ions were sequentially fragmented by higher energy collision dissociation. The MS1 isolation window was set to 1.4 Da and the resolution set at 60,000. MS2 resolution was set at 15,000. The AGC targets were set at $3e^6$ ions for MS1 and $2e^5$ ions for MS2. The normalised collision energy was set at 35%. The maximum ion injection times were set at 50 ms for MS1 and at 19 ms for MS2. The peptides from each fraction were separated using a mix of buffer A (0.1% formic acid in MS grade water) and B (80% acentonitrile, 0.1% formic acid) and eluted from the column using a flow rate of 300 µL/min over 170 min. The column temperature was set at 50 °C with a source voltage of 3 kV. Data were analysed by label-free quantification in MaxQuant (v1.6.1)[70], searching the *T. brucei* L427 reference proteome (version 39, downloaded from TriTrypDB). Calculation of fold-change relative to a sample lacking GFP-tagged VEX2 and statistical analysis were performed using Perseus (v1.5.2.6)[71]. Volcano plots were generated using GraphPad Prism v9.0.

## Co-immunoprecipitation

Bloodstream-form *T. brucei* cells ($4 \times 10^8$) with or without a N-terminal GFP-tagged endogenous copy of *VEX2* and a C-terminal 6xmyc-tagged copy of full-length *VEX1* (native), N-terminal region (native) or C-terminal region (overexpression), were washed three times in ice-cold PBS with EDTA-free protease inhibitors cocktail (Roche) and lysed in ice-cold lysis buffer; RIPA buffer containing 1 mM DTT or 20 mM HEPES, pH 7.4, 1 mM MgCl₂, 10 µM CaCl₂, 100 mM NaCl, 0.1% Tween-20. Both buffers contained protease inhibitor cocktail. Pipetting and incubation for 30 min at 4 °C facilitated lysis in RIPA buffer. Lysates in HEPES buffer were sonicated (8 cycles of 5 s on/5 s off). Samples were spun for 10 min at 20,000 *g* at 4 °C to pellet debris. Supernatant was removed and added to a low protein binding 1.5 mL Eppendorf tube containing α-GFP antibody (Abcam) conjugated to magnetic Dynabeads (Invitrogen) or GFP-Trap magnetic agarose beads (Chromotek), and agitated at 4 °C for 2 h. The samples were then placed on a magnetic rack and washed five times with ice-cold lysis buffer. Samples were eluted with NuPAGE LDS loading buffer containing a reducing agent (Invitrogen). The resulting proteins were fractionated by SDS-PAGE and analysed by protein blotting.

## Native gels

Blue native PAGE was performed using the Native PAGE Bis-Tris gel system (Thermo). Briefly, approximately $8 \times 10^6$ bloodstream-form *T. brucei* cells were washed three times with 1× PBS supplemented with protease inhibitor cocktail without EDTA (Roche), then lysed with 2.5% digitonin in 1× PBS supplemented with protease inhibitor cocktail. Native PAGE sample buffer (Invitrogen) was added to a final concentration of 1× and the samples were then sonicated (8 cycles of 5 s on/5 s off) and spun at 13,000 *g* at 4 °C for 20 min. The resulting supernatants were further treated with micrococcal nuclease (NEB) for 1 h at 37 °C. The samples were split in two: half for the blue native PAGE and the other half for a standard SDS-PAGE (control in denaturing conditions). For the blue native PAGE, Native PAGE sample additive G-250 5% was then added and the samples run on precast 3–12% blue native gradient gels (Invitrogen) at 4 °C according to the

manufacturer's instructions. The gels were submerged in 1% SDS for 30 min with mild agitation prior to transfer onto a PVDF membrane for protein blotting. For the SDS-PAGE, NuPAGE LDS loading buffer + 2.5% β-mercaptoethanol was added. The samples were then denatured for 10 min at 70 °C and run on NuPAGE 4–12% Bis-Tris gradient gels (Invitrogen). As protein standards, NativeMark (unstained) were used for PAGE and PageRuler Plus (stained) were used for SDS-PAGE. The protein equivalent to approximately $2 \times 10^6$ cells was loaded per well.

## Gel filtration
Gel filtration was optimised and performed similarly to refs. [72,73]. Briefly, bloodstream-form T. brucei cells ($1.5 \times 10^8$) were washed three times with 1× PBS supplemented with protease inhibitor cocktail without EDTA, then lysed with 1% CHAPS in 1× PBS supplemented with protease inhibitor cocktail, 1 mM PMSF and 0.5 mM TLCK, followed by mild sonication. Samples were spun at 13,000 g at 4 °C for 20 min and the resulting supernatants were further treated with micrococcal nuclease (NEB) for 1 h at 37 °C. Using a Dionex Ultimate 3000 Bio-RS UHPLC system (Thermo Scientific), the BioBasic SEC 1000 column (Thermo Scientific, 7.8 × 300 mm, 5 µm particles, 1000 Å pores) was washed with 10 volumes of water followed by another 10 volumes of filtered elution buffer (0.3% CHAPS in 1x PBS supplemented with protease inhibitor cocktail, 1 mM PMSF and 0.5 mM TLCK) at 5 °C before the sample injections. 200 µL of filtered samples were injected onto the system at the flow rate of 0.4 mL/min⁻¹ and monitored by a UV detector at 215 nm and 280 nm. The fractions were collected using 96-well Protein Lowbind Deepwell plates (Eppendorf) with well volume of 1000 µL. Thyroglobulin, ferritin, BSA and myoglobulin were used as protein standards. The fractions were then precipitated using trichloroacetic acid following standard procedures, resuspended in NuPAGE LDS loading buffer + 2.5% β-mercaptoethanol, denatured for 10 min at 70 °C and run on NuPAGE 4–12% Bis-Tris gradient gels (Invitrogen) and ultimately analysed by protein blotting.

## Protein blotting
Protein samples were run according to standard protein separation procedures, using SDS-PAGE. However, for VEX2 detection, the use of Bis-Tris gels with a neutral pH environment and a bis–tris/bicine-based transfer buffer (containing a reducing agent and 10% methanol) were critical for protein separation and transfer, respectively (NuPAGE, Invitrogen). Otherwise, protein blotting was carried out according to standard protocols. The following primary antibodies were used: rabbit α-VEX2 (1:1000), rabbit α-VSG-2 (1:20,000), mouse α-myc (Millipore, clone 4A6, 1:7,000), rabbit α-GFP (Abcam, 1:1000), rabbit α-Histone H3 (Abcam, Ab1791) and mouse α-EF1α (Millipore, clone CBP-KK1, 1:20,000). We used horseradish peroxidase coupled secondary antibodies (α-mouse and α-rabbit, Bio-Rad, 1:2000). Blots were developed using an enhanced chemiluminescence kit (Amersham) according to the manufacturer's instructions. Densitometry was performed using Fiji v. 2.0.0[74].

## Immunofluorescence microscopy
Immunofluorescence microscopy was carried out according to standard protocols. For wide-field microscopy, the cells were attached to 12-well 5 mm slides (Thermo Scientific). For super-resolution microscopy, the cells were attached to poly-L-lysine-treated high precision coverslips (thickness 11/2 mm), stained and then mounted onto glass slides. Cells were mounted in Vectashield with DAPI (wide field) or stained with 1 µg mL⁻¹ DAPI for 10 min and then mounted in Vectashield without DAPI (super-resolution). Primary antisera were rat α-VSG-2 (1:10,000), rabbit α-VSG-6 (1:10,000), rabbit α-GFP (Invitrogen, 1:250; Abcam, 1:500), mouse α-myc (NEB, clone 9B11, 1:2000), mouse α-Pol-I (largest subunit; 1:100[20]). The secondary antibodies were Alexa Fluor conjugated goat antibodies: α-mouse, α-rat and α-rabbit, Alexa Fluor

488 or Alexa Fluor 568 (1:1000 for super-resolution microscopy or 1:2000 for wide-field microscopy).

## DNA-fluorescence in-situ hybridisation
For DNA-FISH experiments, biotin- and digoxigenin-labelled DNA probes were generated by PCR using standard conditions with OneTaq polymerase (New England Biolabs), with the exception that a 1:2 ratio of biotin-16-dUTP (Roche) or digoxigenin-11-dUTP (Roche) and dTTP were used in the reaction. Repeats of 50 bp and spliced leader repeats were amplified from T. brucei L427 genomic DNA (primers: P21–P24; Supplementary Data 1, sheet 5). A smear of products was generated but only fragments of 400 bp or less were extracted and purified. DNA probes were co-precipitated with herring sperm DNA (Sigma–Aldrich) at 10 µg/mL and yeast transfer RNA (Invitrogen) at 10 µg/mL. Probes were then resuspended to a concentration of 1000 ng/mL in hybridisation buffer (50% formamide, 10% dextran sulfate and 2× SSC). Before hybridisation, cells were prepared similarly as for immunofluorescence microscopy: trypanosomes were fixed in 3% paraformaldehyde for 15 min at 37 °C, washed three times with PBS and finally resuspended in 1% BSA. The cells were attached to poly-l-lysine-treated slides, then permeabilized with 0.5% Triton X-100 in PBS for 15 min at RT, washed three times with PBS and then treated with 1 mg/mL RNAse A (Invitrogen) in PBS for 1 h at RT. This was followed by a blocking step with 10 µg/mL herring sperm DNA and 10 µg/mL yeast transfer RNA in hybridisation buffer (50% formamide, 10% dextran sulfate and 2× SSC) for 40 min at RT. After adding the probe mix to the slides, the samples were sealed with gene frames and denatured on an inverted heat block at 85 °C for 5 min, followed by O/N incubation at 37 °C. After hybridisation, the slides were washed with 50% formamide and 2× SSC for 30 min at 37 °C, followed by three 10-min washes in 1× SSC, 2× SSC and 4× SSC at 50 °C. Samples were then incubated with an α-digoxigenin antibody (Abcam; clone 21H8) diluted 1:10,000 and rabbit α-myc (NEB; clone 71D10; to stain $^{6myc}$VEX2) diluted 1:500, in 1% BSA in PBS for 2 h at RT. After washing three times for 10 min in Tris-buffered saline with 0.01% Tween-20, the slides were incubated for 1 h with a streptavidin– Alexa Fluor 488 conjugate (Invitrogen), a goat α-mouse Alexa Fluor 568 antibody (Invitrogen) and a goat α-rabbit Alexa Fluor 647 antibody (Invitrogen), all diluted to 1:500 in 1% BSA. Samples were washed in Tris-buffered saline with 0.01% Tween-20, as before, and mounted in Vectashield with DAPI.

## Microscopy and image analysis
For wide-field microscopy, cells were analysed using a Zeiss Axiovert 200 M microscope with an AxioCam MRm camera or a Zeiss AxioObserver Inverted Microscope equipped with Colibri 7 narrow-band LED system and white LED for epifluorescent and white light imaging and ZEN Pro software (Carl Zeiss). Images were acquired as z-stacks (0.1–0.2 µm) and further deconvolved using the default settings ('good, medium') in ZEN Pro. For laser scanning confocal microscopy, images were acquired using a Leica TCS SP8 and Leica Application Suite X software (Leica). For super-resolution microscopy, cells were analysed using a Zeiss LSM880 Airyscan, a Zeiss LSM980 Airyscan 2; or a Zeiss Elyra 7 and the Zeiss ZEN software (Carl Zeiss). Representative images obtained by super-resolution microscopy correspond to maximum 3D projections by the brightest intensity of stacks of approximately 30 slices of 0.1 µm. Images acquired with the Zeiss LSM880 Airyscan were deconvolved using Airyscan SR (XY resolution ~120–140 nm), whereas images acquired with the Zeiss LSM980 Airyscan 2 were deconvolved using Airyscan Joint Deconvolution (XY resolution ~90 nm). Super-resolution structured illumination microscopy (SR-SIM) was performed using a Zeiss Elyra 7 microscope in Lattice SIM[2] mode (XY resolution ~60 nm). SIM reconstruction was performed after correcting for chromatic aberrations using the channel alignment function in Zen and performing deconvolution using the default settings. 100 nm Tetraspeck beads (Thermo) were adhered to

slides and were used to determine channel alignment for each experiment. DAPI-stained *T. brucei* nuclear and mitochondrial DNA were used as cytological markers for cell-cycle stage; one nucleus and one kinetoplast (1N:1K) indicates G1, one nucleus and an elongated kinetoplast (1N:eK) indicates S phase, one nucleus and two kineto-plasts (1N:2K) indicates G2/M and two nuclei and two kinetoplasts (2N:2K) indicates post-mitosis[75,76]. All the images were processed and scored using Fiji v.2.0.0[74]. VEX1 and VEX2 foci and Pol-I nucleolar and expression-site body (ESB) signals could be detected in over 85–90% of nuclei. Prior to the quantification of the distance between N- and C-terminal signals for VEX2, plots showing mean fluorescence intensity (MFI) across the z position for both signals (myc/GFP) were generated and typically the brightest point occurred at the same z coordinate. Therefore, 3D projections of *z*-stacks by brightest intensity were generated and we manually delineated a line profile between each focus of interest using Fiji. The fluorescence intensity output of each channel was then plotted, and the centre of each focus, as determined by the pixel with the greatest intensity, was used to determine the distance. In addition, we measured the 3D distance between both signals by calculating the distance between their centroid positions using the 'Particle Analyser' in Fiji. Pearson's correlation coefficient was applied as a statistical measure of colocalization[77]. Overlapping, adjacent and separate foci presented a Pearson's correlation coefficient in the ranges ≥0.5 to ≤1, ≥−0.5 to <0.5 and ≥−1 to <−0.5, respectively. Counts in total cells were typically performed using >100 nuclei or using >80 nuclei for specific cell cycle phases. All quantifications are averages or representative of at least two biological replicates and independent experiments (see source data for details relating to specific experiments).

### Reporting summary

Further information on research design is available in the Nature Portfolio Reporting Summary linked to this article.

## Data availability

The scRNA-Seq and ChIP-Seq data generated in this study have been deposited in the NCBI database under BioProject accession code PRJNA942067. The LC-MS/MS data generated in this study have been deposited in the ProteomeXchange database under accession code PXD040686. Source Data are provided with this paper.

## Code availability

The code to analyse the scRNA-Seq data has been uploaded in GitHub (https://github.com/mtinti/VSG_single_cell) and deposited in Zenodo (https://doi.org/10.5281/zenodo.10061206).

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

## Acknowledgements

This work was supported by a Wellcome Investigator Award to DH (217105/Z/19/Z) and a Wellcome Trust/Royal Society Sir Henry Dale Fellowship to JFRC (222573/Z/21/Z). We would like to acknowledge the Dundee Imaging Facility, which is supported by a 'Wellcome Technology Platform' award [097945/B/11/Z] and the 'MRC Next Generation Optical Microscopy' award [MR/K015869/1]. We would like to thank the Imaging and Cytometry Lab in the Bioscience Technology Facility at the University of York. We would also like to acknowledge the FingerPrints Proteomics Facility at the University of Dundee, which is supported by a 'Wellcome Trust Technology Platform' award [097945/B/11/Z]. We thank Angus Lamond (University of Dundee) for sharing reagents and granting access to the HPLC equipment for the gel filtration experiments. We thank Rebecca Mitchell (10× Genomics), Emma Briggs (University of Glasgow and University of Edinburgh) and Thomas Otto (University of Glasgow) for advice on single-cell RNA-Seq data analysis. We thank Juan Quintana (University of Glasgow) for advice on native gels. Finally, we acknowledge Lucy Glover (Institut Pasteur) for helpful discussions.

## Author contributions

J.R.C.F. and D.H. conceived the study and planned the experiments. J.R.C.F., C.A.M., H.Y. and M.Z. performed the experiments. J.R.C.F., M.T. and M.Z. analysed the data. M.C.F. provided reagents. J.R.C.F., M.C.F. and D.H. supervised the study. J.R.C.F. and D.H. acquired funding to support the study. J.R.C.F. and D.H. wrote the manuscript, with contributions from all remaining authors.

## Competing interests

The authors declare no competing interests.
