## [Peer Review File · Nature Communications]

An allele-selective inter-chromosomal protein bridge supports monogenic antigen expression in the African trypanosomeReviewers' Comments:

Reviewer #1:

Remarks to the Author:

In short, Faria and Col's manuscript proposes that a VEX1-VEX2 forms a bridge linking two different chromosomes (Chr6 e chr9) to generate multiple allele exclusion. The authors showed that the VEX1-2 complex is self-limiting and that multiple VSGs are expressed in one cell under VEX2 depletion (by scRNA-seq). Although the manuscript is based on solid information, for a non-VSG expert, the results and conclusions seem very similar to what has already been proposed by them and others (see references 13, 18-20 : Faria, J., et al., Spatial integration of transcription and splicing in a dedicated compartment sustains monogenic antigen expression in African trypanosomes. *Nat Microbiol*, 2021. 18. Budzak, J., et al., An assembly of nuclear bodies associates with the active VSG expression site in African trypanosomes. *Nat Commun*, 2022. 19. Faria, J., et al., Monoallelic expression and epigenetic inheritance sustained by a *Trypanosoma brucei* variant surface glycoprotein exclusion complex. *Nat Commun*, 2019.; 20. Glover, L., et al., VEX1 controls the allelic exclusion required for antigenic variation in trypanosomes. *Proc Natl Acad Sci U S A*, 2016.). Some results are undoubtedly original (VEX1-2 complex is self-limiting; multiple VSGs are expressed in one cell under VEX2 depletion, among others), but others are not. For example, one of the main points in this manuscript relies on the fact that VEX1 and VEX2 make a bridge bringing two different chromosomes in close association. However, Vex1 and Vex2 have already been shown to interact with each other before by Faria et al.; 2021. In Faria et al. (2021), "The active-VSG locus and a trans-splicing locus involved in RNA maturation in trypanosomes are also proximal, spatially integrating both transcription and splicing to enhance VSG expression,". Fig 10 from Budzak et al. (2022) has already shown an architecture model of the ES nuclear body, where VEX2 was shown to be at the VSG locus and VEX1 at the VSG active site interacting with each other, in a similar fashion proposed in this current manuscript. Thus, it has already been shown that VEX1 interacts with the SL locus at chr9, and the active VSG is located at chr6 (VSG2). More importantly, VEX1 and VEX2 colocalize. In Extended Data Fig. 7 from Faria et al. (2021- *Nature Microbiology*), they showed "The exclusive association between the active VSG and the SL-locus is VEX2-dependen". In summary, the "bridge" was already evident in previous papers, although the expression "bridge" was first used in the current manuscript. Thus, authors should explicitly explain the main advances shown in their manuscript despite all the similar data already available in the literature.

- Another major issue is regarding the allele-selective role of VEX2. Again, for a non-VSG expert, it is hard to understand these findings based solely on the descriptions of the current manuscript. To be an allele-selective, it must be shown the absence of peaks in the other chr haplotype. In this regard, the authors mentioned that the active VSG-ES is hemizygous. However, different haplotypes of *T. brucei* genomes were already described (Muller et al.; 2018). Is it possible to add different haplotypes (from subtelomeric regions) to the circus plot of Figure 1E? In this figure, the BES and MES genomic regions were removed from individual megabase chr? If not, why an intense VEX2 peak is not found at chr6? Is there any other evidence that VEX2 is an allele-selective?

- "Taken together, these results demonstrated that the VEX2 N- and C-termini extend towards the active-VSG and the SL-RNA compartments, respectively (Extended Data Fig. 4b)." The mentioned figure contains a scheme reflecting microscopic observation from Fig2. However, the abovementioned conclusions are based on indirect observations. To conclude the localization of VEX2 N and C-terminal related to active VSG or SL, authors should perform DNA-FISH assays of these regions. In addition, a zoom-in on Figure 2e e 2f to highlight localization differences should be performed.

Minor

- Figure 1b needs to be clarified. I understood that each "line" represents one ES, and the light grey represents no expression on silent VSGs. Is it possible to add the ES names for each column? Also, the dark grey is so evident that it contrasts with the light grey from the heatmap.

- The Figure 2 scale is incorrect. In fact, there are two bars that do not represent the same size. Please clarify.
- Why not add the extended Fig 4b to the main figures?
- "Derepression appears to follow a well-defined hierarchy: VSG-2 to 6, 15, 17, 18 and finally, VSG-8." Authors should speculate the reasons for that. The discussion section mentioned that it could be due to genomic location. This needs to be clarified for a wider audience not familiar with VSGs. Are these VSGs in the same chr? Are they close to each other?
- scRNA-seq analysis indicated that depletion of VEX2 generates parasites that express two or more VSGs. Is the total VSG transcripts the same as those expressed in wild-type cells? In other words, the sum of each VSG transcript in a VEX2-depleted cell is equivalent to the monoallelically expressed cells?
- "Despite decades of intense study, there remain substantial gaps in our understanding of the mechanisms underpinning gene choice, and singular and switchable antigen expression." It would be nice if authors explicitly cited which are the gaps to understand the monoallelic choice.
- What are the implications of VEX complex being self-limiting? Also, what are the mechanisms for that? Authors should discuss these points.
- Is it known the mechanisms (and signals) that guide VEX2 to select a VSG to be active?
- In the discussion part: the authors mention the importance of long non-coding RNAs and CTCF. The authors should note that a CTCF was not found in trypanosomes.
- What do the pink and green arrows in Fig3c and d mean?

Reviewer #2:

Remarks to the Author:

This excellent paper by Faria et al explores the mechanisms by which monoallelic expression is maintained by the VEX proteins in *Trypanosoma brucei*. Monoallelic expression is important for antigenic variation, which is a main mechanism by which the parasite avoids detection by the host immune system. Thus, deeper insight into the molecular mechanisms that regulate monoallelic expression is essential for understanding the interaction of the parasite with the host immune system, and this insight could be helpful for generating new therapeutic strategies that thwart this immune evasion mechanism. Monoallelic expression is also important in many complex biological systems, such as the olfactory receptor system and the expression of antigen receptors in B and T cells. The study of this process in *T. brucei* may lend insight into similar mechanisms in play for other eukaryotes, and may additionally shed light on how these mechanisms evolved. Thus, the paper should be of interest to both parasitologists and a general audience. In general the experiments are rigorous and well controlled, and the statements made in the paper are well supported by the data. The paper provides data for three main areas that are delineated below. As I read it, the paper makes 3 well supported major points as follows:

1. VEX2 interacts with the active ES. VEX1 binds the spliced leader locus, and the N-terminus of VEX1 interacts with VEX2 to form a bridge between the active ES and the Spliced Leader locus.
2. The VEX2 protein is stabilized by VEX1 and the VEX1 protein is destabilized by VEX2.
3. While wildtype parasites express one VSG per cell, the absence of VEX2 causes 1-8 VSG genes to be expressed within one cell, while the absence of VEX1 results in 1-3 VSG genes expressed within a cell.

Major points.

The authors show data that supports a model wherein the N-terminus of VEX1 interacts with VEX2 to form a bridge between the active ES and the Spliced Leader locus. The authors mentioned that they can endogenously express the C-terminus of VEX1 alone (on page 5, although they mention it is undetectable), I wonder what happens to the bridge under these conditions? My prediction might be that SL arrays would no longer be tethered to VEX2 and the ES, as they were in Figure 1F. This seems to me the key test for the model presented in the paper. Even if the C-terminal VEX1 cannot be detected, the tethering of the ES to the SL array could still be measured using this construct.

In my opinion, the paper could benefit from some streamlining. With 14 extended data figures and 5 main data figures, it's easy to get distracted from the main points the paper is making. If I understood correctly, the main points of the paper are that

1. VEX2 interacts with the active ES. VEX1 binds the spliced leader locus, and the N-terminus of VEX1 interacts with VEX2 to form a bridge between the active ES and the Spliced Leader locus.
2. The VEX2 protein is stabilized by VEX1 and the VEX1 protein is destabilized by VEX2.
3. While wildtype parasites express one VSG per cell, the absence of VEX2 causes 1-8 VSG genes to be expressed within one cell, while the absence of VEX1 results in 1-3 VSG genes expressed within a cell.

The paper would benefit from cutting data from the main figures that does not directly support the three main points above. I've made some suggestions about this below.

Figure 3a and 3b: The authors mention that previous work (Faria et al 2019) has shown that VEX1, VEX2, and CAF-1 form a complex. Given that this has already been shown, this data could be cut or moved to a supplement. There is a point made about telomere proteins here, but it doesn't speak very directly to the main points above.

Figure 3c and 3d. Here, the authors make the point that there is a pool of VEX2 exists independently of VEX1 and that VEX1 may form a dimer. Given that we don't know the residues critical for forming the dimer and thus don't know the biological consequence of having the dimer disrupted, this too could go into a supplement or be cut. I would also suggest that Extended Data Figure 7 be combined with the stability data in Figure 3 so that the effects of VEX1 or VEX2 depletion on protein stability can be directly compared, especially since this is one of the main points made by the paper.

Figure 5. In my view, the key result for VSG expression shown in Figure 5 has been relegated to the supplement (Extended Data 14E). This is a key takeaway and should be in the main figure. It's also difficult to directly compare the effects of knockdown for VEX1 and VEX2 when all the VEX1 data has been put in the supplement. The UMAP projections for the number of expressed VSGs do not convey the result as clearly as the histogram in Extended data 14e and could be removed since they are redundant. The UMAP projections that might be worth keeping are the ones that show the combinations of VSGs, since that data doesn't appear elsewhere. I also didn't find the UMAP projection in 5j very illuminating. In summary I think Figure 5 could be revised to include 5a-f as is, followed by figure 5h, Extended data 14b, and Extended data 14e. The rest could be cut or moved to the supplement.

Extended data Figure 3. This data doesn't seem to add much to this particular story and could be cut.

Extended data Figure 5b. Again, it seems like the existence of the VEX1/VEX2/CAF-1 complex is already known from the Faria et al 2019 paper, so I'm not sure that this is adding much to this particular story and it could be cut.

Extended Data Figure 6. This data could be combined with the data in Figure 3a-d to make one extended figure about complex formation.

Extended Data Figure 11. Simultaneous expression of VSG 2 and VSG-6 has already been shown for VEX2 knockdown in Faria et al 2019. Thus, this figure can be cut.

Extended Data Figure 12 and Figure 13. This data is redundant with the expression profiles shown in Extended Data Figure 10 and can be cut.

General comments.

It might be helpful to revisit the section headings. Some are rather vague and don't make a strong declarative statement about the main result for that section. 'Dissecting interactivity across the VEX1-2 bridge' doesn't really convey the punchline that the N-terminal fragment of VEX1 interacts with VEX2 to form the bridge, for example. Similarly, 'Multi-allelic exclusion' does not convey the main result that depletion of VEX2 results in depression of many VSGs in a single cell, while depletion of VEX1 results in derepression of 2-3 only.

There is a construct that the authors use frequently that makes the results difficult to parse. Example: Both termini can be separately visualized using super-resolution microscopy, with the N- and C-termini extending towards the active-VSG transcription and SL-splicing compartments, respectively. It is much more clear to rewrite the sentence to be

Both termini can be separately visualized using super-resolution microscopy, with the N-terminus extending towards the active-VSG transcription compartment and the C-terminus extending toward the SL-splicing compartment.

The second sentence is much easier to parse. The 'respectively' construct should be used more sparingly, if at all.

Small issues

Figure 1

In the text and methods the authors refer to MACS genome-wide peak calling, but I don't see any indication of where the peaks are called in Figure 1A. The IP/input signal is very strong, so I'm not trying to imply that VEX2 is not localized at the active ES, I just wondered if the algorithm was calling the entire region as one big enriched peak or if it was subdivided into smaller peaks corresponding to each of the genes or areas between genes.

In the text, the authors talk about VEX2 being localized at one corner of chr6 but I don't see that obviously in the figure shown in 1E. Perhaps a box to call our attention to the relevant region would help. Since the authors make a point that the enrichment at the SL locus on chr9 is less pronounced, it would be helpful to show a zoomed in panel for that region underneath the active VSG panel in Figure 1D. Just a note also that VEX2 looks substantially more enriched at the active ES compared to VEX1, but this is hard to see in Figure 1D because the VEX1 data is on a different scale.

Figure 3

In panel, 3C please put in legend what the purple and green arrows mean.

Figure 4

It would be helpful to put in a cartoon of the N and C terminal fragments of VEX1 that are used throughout this figure.

The heading for Figure 4 would be more informative if it made a direct statement of the result, namely that the N-terminus of VEX1 interacts with VEX2.

In Figure 4A, I don't see obvious foci for VEX1, please put arrows to indicate the two foci referred to in the text.

The authors refer to a focus of VEX1 expression when VEX1 is expressed at lower in the 'top panel' of Figure 4b, but it seems to me it is the middle panel that really shows this. I don't see any VEX1 signal in the top panel.

In Figure 4F the authors state that C-terminal VEX1 does not decrease VEX2 abundance, but in the blot that's shown it's difficult to see that because there isn't much VEX2 in the -Tet condition. Given

that this experiment looks like it was done a couple of times, does the other blot show it more clearly? The legend should state what the error on the right panel is showing. I only see 2 dots so it seems unlikely that the error is standard deviation?

Figure 5

It's difficult to make direct comparisons for VSG derepression between wildtype and VEX2 knockdown cells because the violinplots in panel 5B and 5E are not on the same scale. I think there might also be an issue with the legend for Panel i, I don't see a description for these violinplots, though possibly I missed it.

Extended Data 1

For Extended data panels 1D and 1E, can you give more details on the western blot in the legend? It looks like in D you are getting a VEX2 signal in the top panel for clone 1, but not in 1E. Did you use an anti-VEX2 antibody here? Can you be more explicit about what antibody was used for each panel?

Extended Data 4A

Please put labels above blots, Is the bottom blot probed with anti-EF1a etc? The top labels have α next to them but the bottom labels don't.

Reviewer #3:

Remarks to the Author:

This work investigates two proteins identified previously by the authors in greater detail and adds mechanism to the critical role of VEX1 and VEX2 in *T. brucei* antigenic variation. Experiments clearly support the model that VEX1 and VEX2 form a bridge between the active ES and SL array, bind the CAF1 complex and self-regulate. The use of scRNA-seq with the necessary control sample clearly shows loss of mono allelic expression when VEX1/2 are depleted and highlights the preference for ES with two promoter regions. The findings concerning chromatin organisation and this mechanism of allele-selective expression are significant to those in the trypanosome field, as well as those researching antigenic variation and chromatin/gene expression in other organisms.

I have only minor comments for the authors

1. An equivalent plot to 1d for the SL-array would be useful. In 1e there does appear to be a peak in VEX2 binding, but it is unclear if this is the SL-array or an adjacent location. This would help support the claim that VEX2 exclusive binds the active-ES and not the SL array.
2. Extended figure 7b could be moved to main figure to support the key point of the self-regulating complex and reciprocal nature.
3. In figure 4a, the two foci of over expressed full length VEX1 could be labelled for clarity.
4. scRNA-seq showed a greater number of VSGs expressed per cell after VEX2 knockdown compared to VEX1. Could this be due to difference in protein depletion between the two lines at the time point analysed? or do the authors think this is biologically meaningful?

Reviewer #4:

Remarks to the Author:

Faria and colleagues investigate the interplay between VEX2 and VEX1 in the control of allele selective VSG expression in *Trypanosoma brucei* as a strategy of surface antigenic variation for evasion of host immune response. They propose an interesting model to attain monogenic expression through the

formation of VEX2-VEX1 inter-chromosomal bridges between one VSG allele and the SL-arrays. They effectively combine orthogonal techniques such as imaging, biochemical assays and scRNAseq to dissect the molecular basis of monoallelic VSG expression. The work is interesting and well-designed. Still, I have a number of concerns, mainly related to the use and analysis of SR imaging data which I detail below.

Main comments:

1. Regarding the distances measured in 2a/b: I have the following concerns:
 - a. it is unclear how distances between the N and C tags have been measured. Were centroid positions of spots automatically identified? If so, which parameters were used?
 - b. How do the authors explain a shift of 500 nm or more between the N and C terminus of VEX2, which is 224 kDa? Antibodies are approximately 150 kDa and 10-15 nm in size. This is hard to reconcile this with the obtained measures.
 - c. Also, the appearance of two distinguishable spots (one corresponding to Myc, the other to the GFP tag) implies that the entire protein pool should be clustered and perfectly aligned in the same orientation in the nucleus (as the authors indeed propose in the text). How frequently is this arrangement found across imaged nuclei? Is the oligomerization of VEX2 expected to happen in a lateral manner? Is this known from a structural perspective? Can the authors exclude technical artefacts such chromatic aberration or any image reconstruction shift that would favor a shift between magenta and green signals? The authors should provide a detailed explanation to this and a more extended panel of representative figures.
 - d. Lastly, exchanging the position of the two tags is not expected to alter the measured distance so markedly as the VEX2 remains the same size. How do they justify this phenomenon?
 - e. As a general comment regarding imaging datasets, authors should expand the methods section to describe in detail the acquisition and postprocessing conditions as well as analysis steps to improve data reproducibility.
2. In figure 2 it is unclear which datasets have been acquired with confocal microscopy and which with SIM. The "SR" acquisition and post-processing steps of these two approaches is clearly different and not exchangeable or mergeable in the same analysis pipeline. Authors should state clearly which type of imaging is shown for each panel.
3. Figures 3e/, extended 7b/c: were - and + Tet or - and+ MG conditions run in parallel in the same gels? Are the upper and lower WB panels comparable?
4. Figure 4f: authors conclude that VEX2 abundance is not affected by the overexpression of C-terminal VEX1 but the WB bands related to this result show lower abundance and don't match the quantification or the IF results in Fig 4c. Could the authors support this conclusion by showing other WBs with a clear signal?
5. SR images in Fig4g do not look substantially different from diffraction-limited images in Fig4a-c. The VEX1/VEX2 spots look even larger in SR images than in the others. Are the authors performing smoothing, gaussian blurring or any other post-processing modification of the signal? Which is the actual resolution and/or localization precision reached by the SR method used? This is critical as larger/blurrer spots will have higher probabilities to co-localize biasing the results.
6. The authors show how VEX2 and VEX1 bind and form a bridge between a single telomeric VSG-ES and the SL-arrays. Whereas they show that proper levels of both proteins are needed for the presence of the bridge and to maintain monoallelic expression, the relevance of the SL-arrays in this process has not been investigated. The authors could verify the functional relevance of this interaction by investigating the role of RNA and splicing factors in the control of selective VSG expression.
7. Whereas the model proposed implies that the tethering of VEX2 on a specific VSG allele prevents the expression of other VSG genes it doesn't formally demonstrate that the formation of interchromosomal bridge prevents the VEX2 recruitment to other VSG and therefore their derepression. While authors show that depletion of VEX1 leads to derepression of multiple VSG in

single cells they don't show if this is due to the recruitment of VEX2 to these loci or in other words if de-novo binding of VEX2 is sufficient to cause de-repression. A possible way to address this would be to intentionally recruit VEX2 to loci and measure their expression. For example, by fusing VEX2 to dCas9 and recruiting it to multiple VSG genes.

Minor comments:

- In figure panels 2a and 2b the size of the scale bars on the zoomed panels have not been specified in the figure or figure legend. In figure panels 2c 2d the zoomed in panels lack scale bar.
- As a general comment, authors could enlarge imaging panels to improve data visualization.

"An allele-selective inter-chromosomal protein bridge supports monogenic antigen expression in the African trypanosome"

Responses to Reviewer Comments

Reviewer expertise:

Reviewer #1: trypanosome, epigenetics, proteomics

Reviewer #2: trypanosome, VSG expression

Reviewer #3: trypanosome, scRNA-Seq

Reviewer #4: chromatin biology, super-resolution microscopy

Reviewer #1:

1.1: In short, Faria and Col's manuscript proposes that a VEX1-VEX2 forms a bridge linking two different chromosomes (Chr6 e chr9) to generate multiple allele exclusion. The authors showed that the VEX1-2 complex is self-limiting and that multiple VSGs are expressed in one cell under VEX2 depletion (by scRNA-seq). Although the manuscript is based on solid information, for a non-VSG expert, the results and conclusions seem very similar to what has already been proposed by them and others (see references 13, 18-20 : Faria, J., et al., Spatial integration of transcription and splicing in a dedicated compartment sustains monogenic antigen expression in African trypanosomes. Nat Microbiol, 2021. 18. Budzak, J., et al., An assembly of nuclear bodies associates with the active VSG expression site in African trypanosomes. Nat Commun, 2022. 19. Faria, J., et al., Monoallelic expression and epigenetic inheritance sustained by a Trypanosoma brucei variant surface glycoprotein exclusion complex. Nat Commun, 2019.; 20. Glover, L., et al., VEX1 controls the allelic exclusion required for antigenic variation in trypanosomes. Proc Natl Acad Sci U S A, 2016.). Some results are undoubtedly original (VEX1-2 complex is self-limiting; multiple VSGs are expressed in one cell under VEX2 depletion, among others), but others are not. For example, one of the main points in this manuscript relies on the fact that VEX1 and VEX2 make a bridge bringing two different chromosomes in close association. However, Vex1 and Vex2 have already been shown to interact with each other before by Faria et al.; 2021. In Faria et al. (2021), "The active-VSG locus and a trans-splicing locus involved in RNA maturation in trypanosomes are also proximal, spatially integrating both transcription and splicing to enhance VSG expression,". Fig 10 from Budzak et al. (2022) has already shown an architecture model of the ES nuclear body, where VEX2 was shown to be at the VSG locus and VEX1 at the VSG active site interacting with each other, in a similar fashion proposed in this current manuscript. Thus, it has already been shown that VEX1 interacts with the SL locus at chr9, and the active VSG is located at chr6 (VSG2). More importantly, VEX1 and VEX2 colocalize. In Extended Data Fig. 7 from Faria et al. (2021- Nature Microbiology), they showed "The exclusive association between the active VSG and the SL-locus is VEX2-dependen". In summary, the "bridge" was already evident in previous papers, although the expression "bridge" was first used in the current manuscript. Thus, authors should explicitly explain the main advances shown in their manuscript despite all the similar data already available in the literature.

R1.1: Prior data were certainly consistent with a 'bridge', but colocalization by microscopy does not demonstrate a physical interaction between VEX proteins and chromosomes, while Chromosome Conformation Capture can reveal proximity and frequent interactions or "collisions", but again, not a protein bridge. An allele-selective interaction uncovered by ChIP-Seq, as shown in Fig. 1, and an inter-chromosomal protein bridge, as elaborated in subsequent Figures, are unprecedented, as far as we are aware. Having checked our Abstract, we feel that this does explicitly explain the main advances shown in the manuscript, although we're happy to take advice if this remains unclear.

1.2: Another major issue is regarding the allele-selective role of VEX2. Again, for a non-VSG expert, it is hard to understand these findings based solely on the descriptions of the current

manuscript. To be an allele-selective, it must be shown the absence of peaks in the other chr haplotype. In this regard, the authors mentioned that the active VSG-ES is hemizygous. However, different haplotypes of *T. brucei* genomes were already described (Muller et al.; 2018). Is it possible to add different haplotypes (from subtelomeric regions) to the circos plot of Figure 1E? In this figure, the BES and MES genomic regions were removed from individual megabase chr? If not, why an intense VEX2 peak is not found at chr6? Is there any other evidence that VEX2 is an allele-selective?

R1.2: We've added some text to clarify in the Results section: "A circos plot reveals this signal specifically in the strain expressing affinity-tagged VEX2 and specifically at the active VSG expression site in that strain (Fig. 1a); note that BES1 has been mapped to chromosome 6 [Muller et al., 2018] but because genome sequencing projects for *T. brucei* have not yet delivered telomere-to-telomere chromosome assemblies, the telomeric expression sites [Hertz-Fowler et al., 2008; Kolev et al., 2012; Cross et al., 2014] are shown separately in the circos plot." We used the reference genome assembly to generate the circos plot in Fig. 1a and have now added the following "We also mapped the VEX2 ChIP-Seq data to a second genome assembly [Muller et al., 2018] that includes hemizygous sub-telomeric sequences (Fig. S2d). As above, this circos plot reveals VEX2 chromatin binding specifically in the strain expressing affinity-tagged VEX2 and specifically at the active VSG expression site in that strain".

1.3: "Taken together, these results demonstrated that the VEX2 N- and C-termini extend towards the active-VSG and the SL-RNA compartments, respectively (Extended Data Fig. 4b)." The mentioned figure contains a scheme reflecting microscopic observation from Fig2. However, the abovementioned conclusions are based on indirect observations. To conclude the localization of VEX2 N and C-terminal related to active VSG or SL, authors should perform DNA-FISH assays of these regions. In addition, a zoom-in on Figure 2e e 2f to highlight localization differences should be performed.

R1.3: We've adjusted the text here such that our conclusions are now based on direct observations; "Taken together, these results indicated that the VEX2 N-terminus extends towards the extranucleolar Pol-I compartment while the VEX2 C-terminus extends towards a VEX1 compartment". The following text has also been adjusted to further clarify "Since VEX1 is known to bind SL-RNA arrays (Faria et al., 2021), and to form a complex with VEX2 (Faria et al., 2019), which is now known to bind the active VSG-ES (Fig. 1), we conclude that the VEX1-VEX2 complex bridges two chromosomes, linking the active VSG-ES to a major RNA splicing locus". The underlined text has been added. We have also now included additional zoomed images in Fig. 2 as suggested.

Minor

1.4: Figure 1b needs to be clarified. I understood that each "line" represents one ES, and the light grey represents no expression on silent VSGs. Is it possible to add the ES names for each column? Also, the dark grey is so evident that it contrasts with the light grey from the heatmap.

R1.4: Since this ChIP-Seq panel is a representation of a subset of the data shown in the circos plot (Fig. 1a), and Fig. S2b already shows data for all VSG-ESs, we moved this panel to Fig. S2c, and also adjusted the dark grey region.

1.5: The Figure 2 scale is incorrect. In fact, there are two bars that do not represent the same size. Please clarify.

R1.5: Apologies for the confusion. These were not scale bars, the lines indicated the regions analysed for the histogram (now Fig. 2a only). These lines have now been removed.

1.6: Why not add the extended Fig 4b to the main figures?

R1.6: Thank you for the suggestion – panel now included in Fig. 5c.

1.7: "Derepression appears to follow a well-defined hierarchy: VSG-2 to 6, 15, 17, 18 and finally, VSG-8". Authors should speculate the reasons for that. The discussion section mentioned that it could be due to genomic location. This needs to be clarified for a wider

audience not familiar with VSGs. Are these VSGs in the same chr? Are they close to each other?

R1.7: Good point. We've now added "nuclear positioning, or accessibility of the telomeric sites on several distinct chromosomes where these VSGs are located; the VSG-2, 6, 18 and 8 expression sites are on megabase chromosomes, while the VSG-15 and 17 expression sites are on intermediate chromosomes".

1.8: scRNA-seq analysis indicated that depletion of VEX2 generates parasites that express two or more VSGs. Is the total VSG transcripts the same as those expressed in wild-type cells? In other words, the sum of each VSG transcript in a VEX2-depleted cell is equivalent to the monoallelically expressed cells?

R1.8: An interesting question. We've added a box plot, and the following text: "Notably, following VEX2 knockdown, total VSG levels are substantially increased in cells expressing multiple VSGs, while VSG-2 expression itself is not substantially altered in these cells (Fig. 7e)".

1.9: "Despite decades of intense study, there remain substantial gaps in our understanding of the mechanisms underpinning gene choice, and singular and switchable antigen expression." It would be nice if authors explicitly cited which are the gaps to understand the monoallelic choice.

R1.9: This text in the Discussion follows related text in the Introduction: "Despite decades of intense study, the molecular mechanisms facilitating the selection and maintenance of expression of a single active allele, coordinated with silencing of all others, and stochastic, low-frequency transcriptional switching, remain poorly understood". The underlined text has now been added.

1.10: What are the implications of VEX complex being self-limiting? Also, what are the mechanisms for that? Authors should discuss these points.

R1.10: We've added the following text to the Discussion: "turnover of both proteins is proteasome-dependent. Notably, native VEX1 stabilises VEX2, while overexpressed VEX1 destabilises VEX2. We suggest that VEX1 selectively stabilises VEX2 in the context of an active VSG expression site. Thus, intrinsically self-limiting control of the VEX-complex could facilitate allele selective chromosomal bridging and limit the pool of VEX2 available to accumulate at other VSG expression sites".

1.11: Is it known the mechanisms (and signals) that guide VEX2 to select a VSG to be active?

R1.11: This appears to be stochastic. "VSG expression site promoters drive the competition for activation" (Discussion), while "VEX1 and VEX2 assemble in an RNA polymerase-I transcription-dependent manner" (Faria *et al.*, 2019).

1.12: In the discussion part: the authors mention the importance of long non-coding RNAs and CTCF. The authors should note that a CTCF was not found in trypanosomes.

R1.12: We'd prefer to reserve judgement on this point. Notably, Tb927.9.2780 (hypothetical protein, conserved) is (weakly) related to CTCF.

1.13: What do the pink and green arrows in Fig3c and d mean?

R1.13: These panels are now Fig. 3b-c. The arrows were included to indicate "slower-migrating forms diminished following VEX2 depletion" (Fig. 3b) and "a distribution that was maintained following VEX1 ... depletion" (Fig. 3c), but these have now been removed as the data and main text suffice to explain these points.

Reviewer #2:

This excellent paper by Faria et al explores the mechanisms by which monoallelic expression is maintained by the VEX proteins in *Trypanosoma brucei*. Monoallelic expression is important for antigenic variation, which is a main mechanism by which the parasite avoids detection by the host immune system. Thus, deeper insight into the molecular mechanisms that regulate monoallelic expression is essential for understanding the interaction of the parasite with the host immune system, and this insight could be helpful for generating new therapeutic strategies that thwart this immune evasion mechanism. Monoallelic expression is also important in many complex biological systems, such as the olfactory receptor system and the

expression of antigen receptors in B and T cells. The study of this process in *T. brucei* may lend insight into similar mechanisms in play for other eukaryotes, and may additionally shed light on how these mechanisms evolved. Thus, the paper should be of interest to both parasitologists and a general audience. In general the experiments are rigorous and well controlled, and the statements made in the paper are well supported by the data. The paper provides data for three main areas that are delineated below. As I read it, the paper makes 3 well supported major points as follows:

1. VEX2 interacts with the active ES. VEX1 binds the spliced leader locus, and the N-terminus of VEX1 interacts with VEX2 to form a bridge between the active ES and the Spliced Leader locus.
2. The VEX2 protein is stabilized by VEX1 and the VEX1 protein is destabilized by VEX2.
3. While wildtype parasites express one VSG per cell, the absence of VEX2 causes 1-8 VSG genes to be expressed within one cell, while the absence of VEX1 results in 1-3 VSG genes expressed within a cell.

Major points.

2.1: The authors show data that supports a model wherein the N-terminus of VEX1 interacts with VEX2 to form a bridge between the active ES and the Spliced Leader locus. The authors mentioned that they can endogenously express the C-terminus of VEX1 alone (on page 5, although they mention it is undetectable), I wonder what happens to the bridge under these conditions? My prediction might be that SL arrays would no longer be tethered to VEX2 and the ES, as they were in Figure 1F. This seems to me the key test for the model presented in the paper. Even if the C-terminal VEX1 cannot be detected, the tethering of the ES to the SL array could still be measured using this construct.

R2.1: An unmodified allele of VEX1 is still present in cells expressing this C-terminal fragment of VEX1. We have been unable to disrupt both alleles of VEX1, which suggests that VEX1 function is required for viability.

2.2: In my opinion, the paper could benefit from some streamlining. With 14 extended data figures and 5 main data figures, it's easy to get distracted from the main points the paper is making. If I understood correctly, the main points of the paper are that

1. VEX2 interacts with the active ES. VEX1 binds the spliced leader locus, and the N-terminus of VEX1 interacts with VEX2 to form a bridge between the active ES and the Spliced Leader locus.
2. The VEX2 protein is stabilized by VEX1 and the VEX1 protein is destabilized by VEX2.
3. While wildtype parasites express one VSG per cell, the absence of VEX2 causes 1-8 VSG genes to be expressed within one cell, while the absence of VEX1 results in 1-3 VSG genes expressed within a cell.

The paper would benefit from cutting data from the main figures that does not directly support the three main points above. I've made some suggestions about this below.

R2.2: Thank-you for this point. Instead of 5 main Figures and 14 Supplementary Figures, the manuscript now has 8 main Figures and 8 Supplementary Figures. Further details below.

2.2a: Figure 3a and 3b: The authors mention that previous work (Faria et al 2019) has shown that VEX1, VEX2, and CAF-1 form a complex. Given that this has already been shown, this data could be cut or moved to a supplement. There is a point made about telomere proteins here, but it doesn't speak very directly to the main points above.

R2.2a: The Fig. 3a panel has now been moved to Fig. S4b. We've retained one of the immunoprecipitation panels here (Fig. 3a), however, as this shows that VEX2 primarily interacts with VEX1 and CAF-1, but not with (an)other protein(s). This is a new conclusion derived from the current data.

2.2b: Figure 3c and 3d. Here, the authors make the point that there is a pool of VEX2 exists independently of VEX1 and that VEX1 may form a dimer. Given that we don't know the

residues critical for forming the dimer and thus don't know the biological consequence of having the dimer disrupted, this too could go into a supplement or be cut. I would also suggest that Extended Data Figure 7 be combined with the stability data in Figure 3 so that the effects of VEX1 or VEX2 depletion on protein stability can be directly compared, especially since this is one of the main points made by the paper.

R2.2b: These panels are now Fig. 3b-c; the key point here, that we prefer to retain in a main Figure, is that VEX2 increases the size of VEX1-containing complexes. We've moved data showing VEX2-dependent VEX1 abundance from Fig. S7b to Fig. 3d, as suggested.

2.2c: Figure 5. In my view, the key result for VSG expression shown in Figure 5 has been relegated to the supplement (Extended Data 14E). This is a key takeaway and should be in the main figure. It's also difficult to directly compare the effects of knockdown for VEX1 and VEX2 when all the VEX1 data has been put in the supplement. The UMAP projections for the number of expressed VSGs do not convey the result as clearly as the histogram in Extended data 14e and could be removed since they are redundant. The UMAP projections that might be worth keeping are the ones that show the combinations of VSGs, since that data doesn't appear elsewhere. I also didn't find the UMAP projection in 5j very illuminating. In summary I think Figure 5 could be revised to include 5a-f as is, followed by figure 5h, Extended data 14b, and Extended data 14e. The rest could be cut or moved to the supplement.

R2.2c: The data from Fig. S14b and S14e are now shown in a main Figure as suggested; new Fig. 8b and 8e. We have retained prior Fig. 5g, i and j panels, however, now Fig. 7a, c and d as we feel that these panels convey important information, as detailed in the text.

2.2d: Extended data Figure 3. This data doesn't seem to add much to this particular story and could be cut.

R2.2d: These data, relating to chromatin-bound and soluble fractions of VEX1, VEX2 and CAF-1 have been removed.

2.2e: Extended data Figure 5b. Again, it seems like the existence of the VEX1/VEX2/CAF-1 complex is already known from the Faria et al 2019 paper, so I'm not sure that this is adding much to this particular story and it could be cut.

R2.2e: We have retained these data panels (Fig. S4c-d) as they provide what we believe is important information relating to the sub-nuclear location of the VEX-complex relative to Pol-I, Pol-II and nucleolar compartments.

2.2f Extended Data Figure 6. This data could be combined with the data in Figure 3a-d to make one extended figure about complex formation.

R2.2f: See R2.2b above.

2.2g: Extended Data Figure 11. Simultaneous expression of VSG 2 and VSG-6 has already been shown for VEX2 knockdown in Faria et al 2019. Thus, this figure can be cut.

R2.2g: This Figure has been removed as suggested.

2.2h: Extended Data Figure 12 and Figure 13. This data is redundant with the expression profiles shown in Extended Data Figure 10 and can be cut.

R2.2h: These data are not redundant in the sense that Fig. S10 showed expression data for individual VSGs, while Fig. S12-13 showed multi-VSG expression data for individual cells, as well as illustrating that our biological replicates yielded similar results. Data from prior Fig's. S10 and S12-13 have now been combined in Fig. S8.

General comments.

2.3: It might be helpful to revisit the section headings. Some are rather vague and don't make a strong declarative statement about the main result for that section. 'Dissecting interactivity across the VEX1-2 bridge' doesn't really convey the punchline that the N-terminal fragment of VEX1 interacts with VEX2 to form the bridge, for example. Similarly, 'Multi-allelic exclusion'

does not convey the main result that depletion of VEX2 results in depression of many VSGs in a single cell, while depletion of VEX1 results in derepression of 2-3 only.

R2.3: Thank-you for the suggestions here. We have changed the first sub-heading to “The N-terminal fragment of VEX1 interacts with VEX2 to form the bridge”, as suggested. We’ve also added further text to summarise the findings at the end of this section: “a VEX2-interacting and probable bridging domain resides within the N-terminal region of VEX1. The C-terminal region of VEX1 contains an NLS, and the C-terminal fragment of VEX1 analysed here also increases turnover.” The second sub-heading has also been adjusted as suggested; “VEX2 maintains multi-allelic exclusion”.

2.4: There is a construct that the authors use frequently that makes the results difficult to parse. Example: Both termini can be separately visualized using super-resolution microscopy, with the N- and C- termini extending towards the active-VSG transcription and SL-splicing compartments, respectively. It is much more clear to rewrite the sentence to be

Both termini can be separately visualized using super-resolution microscopy, with the N-terminus extending towards the active-VSG transcription compartment and the C-terminus extending toward the SL-splicing compartment.

The second sentence is much easier to parse. The ‘respectively’ construct should be used more sparingly, if at all.

R2.4: Good point. We had five instances of ‘respectively’ in the Results section and three in the Discussion. All but one of these in the Discussion have been removed.

Small issues

2.5: Figure 1. In the text and methods the authors refer to MACS genome-wide peak calling, but I don’t see any indication of where the peaks are called in Figure 1A. The IP/input signal is very strong, so I’m not trying to imply that VEX2 is not localized at the active ES, I just wondered if the algorithm was calling the entire region as one big enriched peak or if it was subdivided into smaller peaks corresponding to each of the genes or areas between genes.

R2.5: MACS2 calls multiple peaks that include CDSs, CDSs + intergenic regions or intergenic regions. We’ve added the peak-calling coordinates to Supplementary Data Sheet 1.

2.6: In the text, the authors talk about VEX2 being localized at one corner of chr6 but I don’t see that obviously in the figure shown in 1E. Perhaps a box to call our attention to the relevant region would help. Since the authors make a point that the enrichment at the SL locus on chr9 is less pronounced, it would be helpful to show a zoomed in panel for that region underneath the active VSG panel in Figure 1D. Just a note also that VEX2 looks substantially more enriched at the active ES compared to VEX1, but this is hard to see in Figure 1D because the VEX1 data is on a different scale.

R2.6: This relates to response R1.2 above. We’ve added some text to clarify in the Results section: “A circos plot reveals this signal specifically in the strain expressing affinity-tagged VEX2 and specifically at the active VSG expression site in that strain (Fig. 1a); note that BES1 has been mapped to chromosome 6 [Muller *et al.*, 2018] but because genome sequencing projects for *T. brucei* have not yet delivered telomere-to-telomere chromosome assemblies, the telomeric expression sites [Hertz-Fowler *et al.*, 2008 / Kolev *et al.*, 2012 / Cross *et al.*, 2014] are shown separately in the circos plot. To address these other points, Fig. 1d now shows a comparison of VEX1 and VEX2 ChIP-Seq signals at the SL locus, and Fig. 1c now shows VEX1 and VEX2 enrichment on the same scale. To support the former point, we added the text: “indeed, the SL-array was strongly enriched following VEX1 ChIP-Seq (Fig. 1d), as determined previously [Faria *et al.*, 2019]”.

2.7: Figure 3. In panel, 3C please put in legend what the purple and green arrows mean.

R2.7: see R1.13 above.

2.8: Figure 4. It would be helpful to put in a cartoon of the N and C terminal fragments of VEX1 that are used throughout this figure.

R2.8: Now included as suggested (Fig. 4a).

2.9: The heading for Figure 4 would be more informative if it made a direct statement of the result, namely that the N-terminus of VEX1 interacts with VEX2.

R2.9: Some of these data now appear in Fig. 5, with the suggested heading, “The N-terminal region of VEX1 interacts with VEX2”. Also see R2.3 above.

2.10: In Figure 4A, I don't see obvious foci for VEX1, please put arrows to indicate the two foci referred to in the text.

R2.10: We have added arrows indicating “regions that displayed a more intense signal” (now Fig. 4b).

2.11: The authors refer to a focus of VEX1 expression when VEX1 is expressed at lower in the ‘top panel’ of Figure 4b, but it seems to me it is the middle panel that really shows this. I don't see any VEX1 signal in the top panel.

R2.11: Thank-you for spotting this error - “top” has been adjusted to “middle”.

2.12: In Figure 4F the authors state that C-terminal VEX1 does not decrease VEX2 abundance, but in the blot that's shown it's difficult to see that because there isn't much VEX2 in the -Tet condition. Given that this experiment looks like it was done a couple of times, does the other blot show it more clearly? The legend should state what the error on the right panel is showing. I only see 2 dots so it seems unlikely that the error is standard deviation?

R2.12: We've now shown the second blot here (now Fig. 4g), and both uncropped blots are included in the ‘Source Data’ file. The error bars do indeed indicate standard deviation, and this is now stated in the legend.

2.13: Figure 5. It's difficult to make direct comparisons for VSG derepression between wildtype and VEX2 knockdown cells because the violinplots in panel 5B and 5E are not on the same scale. I think there might also be an issue with the legend for Panel i, I don't see a description for these violinplots, though possibly I missed it.

R2.13: Good point – these violin plots (and an equivalent for VEX1 knockdown) are now in Figure panels 6d-f and are all now on the same scale. There was indeed an issue with the legend for panel i (the box-plot) – thank-you for spotting this also. This has been corrected – now Fig. 7c.

2.14: Extended Data 1

For Extended data panels 1D and 1E, can you give more details on the western blot in the legend? It looks like in D you are getting a VEX2 signal in the top panel for clone 1, but not in 1E. Did you use an anti-VEX2 antibody here? Can you be more explicit about what antibody was used for each panel?

R2.14: We've added more detail to the Figure S1d-e panels, and the following text to the legend, to clarify; “both α -VEX2 (top panels) and α -myc (middle panels) were used”.

2.15: Extended Data 4A. Please put labels above blots, Is the bottom blot probed with anti-EF1a etc? The top labels have α next to them but the bottom labels don't.

R2.15: α added, now S1h.

Reviewer #3:

This work investigates two proteins identified previously by the authors in greater detail and adds mechanism to the critical role of VEX1 and VEX2 in *T. brucei* antigenic variation.

Experiments clearly support the model that VEX1 and VEX2 form a bridge between the active ES and SL array, bind the CAF1 complex and self-regulate. The use of scRNA-seq with the necessary control sample clearly shows loss of mono allelic expression when VEX1/2 are depleted and highlights the preference for ES with two promoter regions. The findings concerning chromatin organisation and this mechanism of allele-selective expression are significant to those in the trypanosome field, as well as those researching antigenic variation and chromatin/gene expression in other organisms.

I have only minor comments for the authors

3.1: An equivalent plot to 1d for the SL-array would be useful. In 1e there does appear to be a peak in VEX2 binding, but it is unclear if this is the SL-array or an adjacent location. This would help support the claim that VEX2 exclusive binds the active-ES and not the SL array.

R3.1: Fig. 1d now shows a comparison of VEX1 and VEX2 ChIP-Seq signals at the SL locus.

3.2: Extended figure 7b could be moved to main figure to support the key point of the self-regulating complex and reciprocal nature.

R3.2: We've moved data showing VEX2-dependent VEX1 abundance from Fig. S7b to Fig. 3d, as suggested.

3.3: In figure 4a, the two foci of over expressed full length VEX1 could be labelled for clarity.

R3.3: See R2.10.

3.4: scRNA-seq showed a greater number of VSGs expressed per cell after VEX2 knockdown compared to VEX1. Could this be due to difference in protein depletion between the two lines at the time point analysed? or do the authors think this is biologically meaningful?

R3.4: Our view is that this is biologically meaningful, since we have no reason to believe that the penetrance of RNAi knockdown is substantially different for VEX1 or VEX2 (see Fig. 5f in Faria *et al.*, 2019). Although VEX1 and VEX2 function in a complex, and each protein has an impact on the abundance of the other, the allele-selective protein, VEX2, appears to function as the primary 'exclusion factor'.

Reviewer #4:

Faria and colleagues investigate the interplay between VEX2 and VEX1 in the control of allele selective VSG expression in *Trypanosoma brucei* as a strategy of surface antigenic variation for evasion of host immune response. They propose an interesting model to attain monogenic expression through the formation of VEX2-VEX1 inter-chromosomal bridges between one VSG allele and the SL-arrays. They effectively combine orthogonal techniques such as imaging, biochemical assays and scRNAseq to dissect the molecular basis of monoallelic VSG expression. The work is interesting and well-designed. Still, I have a number of concerns, mainly related to the use and analysis of SR imaging data which I detail below.

Main comments:

Regarding the distances measured in 2a/b: I have the following concerns:

4.1a: it is unclear how distances between the N and C tags have been measured. Were centroid positions of spots automatically identified? If so, which parameters were used?

R4.1a: We've added further detail in the Methods section (Microscopy and image analysis): "Quantification of the distance between N and C-terminal signals for VEX2 was performed by generating 3D projections of z-stacks by brightest intensity and manually delineating a line profile between each focus of interest using Fiji. The fluorescence intensity output of each channel was then plotted, and the centre of each focus, as determined by the pixel with the greatest intensity, was used to determine the distance".

4.1b: How do the authors explain a shift of 500 nm or more between the N and C terminus of VEX2, which is 224 kDa? Antibodies are approximately 150 kDa and 10-15 nm in size. This is hard to reconcile this with the obtained measures.

R4.1b: Thank you for raising this point, which revealed an error in one of the histograms. We have now quantified VEX2 signal diameter in many additional cells and present these data in a violin plot (Fig. 2c); details have been added to the Figure legend.

4.1c: Also, the appearance of two distinguishable spots (one corresponding to Myc, the other to the GFP tag) implies that the entire protein pool should be clustered and perfectly aligned in the same orientation in the nucleus (as the authors indeed propose in the text). How frequently is this arrangement found across imaged nuclei? Is the oligomerization of VEX2 expected to happen in a lateral manner? Is this known from a structural perspective? Can the authors exclude technical artefacts such chromatic aberration or any image reconstruction shift that would favor a shift between magenta and green signals? The authors should provide a detailed explanation to this and a more extended panel of representative figures.

R4.1c: See R1.4b above. Oligomerisation in a lateral manner was neither expected nor known for VEX2, but our data do suggest “that a pool of VEX2 protein displays a common orientation at this site”.

4.1d: Lastly, exchanging the position of the two tags is not expected to alter the measured distance so markedly as the VEX2 remains the same size. How do they justify this phenomenon?

R4.1d: See R4.1b above. Quantification (Fig. 2c) reveals that the measured distances are not markedly altered when the two tags are exchanged.

4.1e: As a general comment regarding imaging datasets, authors should expand the methods section to describe in detail the acquisition and postprocessing conditions as well as analysis steps to improve data reproducibility.

R4.1e: We've added further detail in the Methods section (Microscopy and image analysis), see R4.1a above and R4.5 below.

4.2: In figure 2 it is unclear which datasets have been acquired with confocal microscopy and which with SIM. The “SR” acquisition and post-processing steps of these two approaches is clearly different and not exchangeable or mergeable in the same analysis pipeline. Authors should state clearly which type of imaging is shown for each panel.

R4.2: We have clarified these aspects both in the Methods section (Microscopy and image analysis), and in the Figure legends. Also see R4.5 below.

4.3: Figures 3e/, extended 7b/c: were – and + Tet or – and+ MG conditions run in parallel in the same gels? Are the upper and lower WB panels comparable?

R4.3: They are indeed comparable. We now state in the legends (now Fig. 3d-f); “EF1 α was used as a loading control on the same blots used for VEX1 or VEX2 analysis; -Tet/+Tet or -MG132/+MG132 samples were run in the same gel for each experiment”. Uncropped blots are included in the Source Data File.

4.4: Figure 4f: authors conclude that VEX2 abundance is not affected by the overexpression of C-terminal VEX1 but the WB bands related to this result show lower abundance and don't match the quantification or the IF results in Fig 4c. Could the authors support this conclusion by showing other WBs with a clear signal?

R4.4: See R2.12 above.

4.5: SR images in Fig4g do not look substantially different from diffraction-limited images in Fig4a-c. The VEX1/VEX2 spots look even larger in SR images than in the others. Are the authors performing smoothing, gaussian blurring or any other post-processing modification of the signal? Which is the actual resolution and/or localization precision reached by the SR

method used? This is critical as larger/blurrier spots will have higher probabilities to co-localize biasing the results.

R4.5: These images are now in Fig. 5a and Fig. 4b-d. We used microscopes with different XY resolution. Details have now been added to the Methods section (Microscopy and image analysis) and we have now indicated which microscope was used to acquire the images in each Figure legend.

4.6: The authors show how VEX2 and VEX1 bind and form a bridge between a single telomeric VSG-ES and the SL-arrays. Whereas they show that proper levels of both proteins are needed for the presence of the bridge and to maintain monoallelic expression, the relevance of the SL-arrays in this process has not been investigated. The authors could verify the functional relevance of this interaction by investigating the role of RNA and splicing factors in the control of selective VSG expression.

R4.6: We investigated the role of Spliced Leader Array Protein 1 (SLAP1, Tb927.11.9950) in VEX2 sequestration and have now included these data (Fig. S3). The results are “consistent with the view that VEX2 sequestration is dependent upon mRNA *trans*-splicing”.

4.7: Whereas the model proposed implies that the tethering of VEX2 on a specific VSG allele prevents the expression of other VSG genes it doesn't formally demonstrate that the formation of interchromosomal bridge prevents the VEX2 recruitment to other VSG and therefore their derepression. While authors show that depletion of VEX1 leads to derepression of multiple VSG in single cells they don't show if this is due to the recruitment of VEX2 to these loci or in other words if de-novo binding of VEX2 is sufficient to cause de-repression. A possible way to address this would be to intentionally recruit VEX2 to loci and measure their expression. For example, by fusing VEX2 to dCas9 and recruiting it to multiple VSG genes.

R4.7: We have considered experiments of this kind. Unfortunately, this is beyond the scope of the current study. First, because the VEX2 CDS is 6.1 kbp and has proven challenging to express ectopically; inducible expression would likely be required for this experiment. Second, because the dCas9 technology has not yet been effectively developed / applied to *T. brucei* studies, as far as we are aware. It is also possible that direct tethering to DNA will interfere with VSG expression site transcription.

Minor comments:

4.8: In figure panels 2a and 2b the size of the scale bars on the zoomed panels have not been specified in the figure or figure legend. In figure panels 2c 2d the zoomed in panels lack scale bar.

R4.8: See R1.5 above.

4.9: As a general comment, authors could enlarge imaging panels to improve data visualization.

R4.9: We have now included seven enlarged and zoomed images in Fig. 2.

Reviewers' Comments:

Reviewer #1:

Remarks to the Author:

The authors have now answered most of the concerns about their data. Some clarifications on the text and the removal of figures that showed previously published results improved the manuscript. The inclusion of additional figures suggested by me, and others, were also considered. No further revisions are required.

Reviewer #2:

Remarks to the Author:

I appreciate that the authors addressed all of my previous concerns in this revised version of the manuscript, and I would support publication of the paper in its current form.

Reviewer #3:

Remarks to the Author:

The revised manuscript from Faria et al clarified the points raised by reviewers. Although this work does build on that of previous publications, the experiments presented here are new.

Previous work has shown VEX1 and VEX2 knockdown result in the transcription of previously silent ES at a population level, and the simultaneous expression of two VSGs on the surface of the same parasites. This is manuscript, Faria et al show that a single parasite can in fact transcribe up to 12 ESs at the same time after VEX2 depletion, and that the originally active VSG is expressed at the same level.

Previously work by the authors demonstrated that VEX1 binds the SL array, that VEX1 and VEX2 interact and colocalise in the nucleus, and that the SL and active ES are in close proximity. They now demonstrate that VEX2 directs binds to the active ES, and that it is located across the ES rather than one location, for example. They also investigate the fragments of proteins required for VEX1-VEX2 interactions and demonstrate that the VEX1-2 bridge is self-limiting.

Importantly, these findings are in keeping with the wider model of the proteins present within different compartments at the ES nuclear body described by Budzak et al. In this revised version of the manuscript Faria et al now show that VEX2 sequestration to the active ES is dependent on SLAP1, a key protein required for SL-RNA production, and ES transcription, identified by Budzak et al.

I have only one outstanding point:

The authors need to provide all the code used in their analysis. In particular, the custom script they now describe in the material and methods section (line 513-523).

Reviewer #4:

Remarks to the Author:

The authors have addressed most of my comments. However, the following points still require further clarification:

Comment r4.1.a: instead of measuring real 3D distances (Euclidean distances) the authors reduce the dimensionality information to 2D by making 2D projections, this obviously affects the values obtained. For example, two objects localized in the same x-y coordinates but separated by a significant distance (even meters) in z will perfectly colocalize in 2D when a z projection is made. The type of

quantification done by the authors is therefore not appropriate.

Comment r4.1.b: since the histogram was not representative for panel 2b please remove it also from Fig 2a, otherwise it is misleading to keep only the one that shows more vicinity. A quantification such as the one added in Fig2c is more informative.

Comment r4.1.c: the reply is not conclusive.

An allele-selective inter-chromosomal protein bridge supports monogenic antigen expression in the African trypanosome

Responses to further reviewer comments:

Reviewer #3:

I have only one outstanding point:

The authors need to provide all the code used in their analysis. In particular, the custom script they now describe in the material and methods section (line 513-523).

The code to analyse the single cell data has been uploaded to GitHub (https://github.com/mtinti/VSG_single_cell) and Zenodo DOI: 10.5281/zenodo.10061206. We've added a 'Code Availability' section.

Reviewer #4:

The authors have addressed most of my comments. However, the following points still require further clarification:

Comment r4.1.a: instead of measuring real 3D distances (Euclidean distances) the authors reduce the dimensionality information to 2D by making 2D projections, this obviously affects the values obtained. For example, two objects localized in the same x-y coordinates but separated by a significant distance (even meters) in z will perfectly colocalize in 2D when a z projection is made. The type of quantification done by the authors is therefore not appropriate.

Response to comment r4.1a: We have now calculated 3D distances by determining the distance between centroid points (using the 'Particle Analyser' in Fiji) – see the new violin plot on the right hand-side in Fig. 2c, and the modified text in the materials and methods and Fig. 2 legend.

Comment r4.1.b: since the histogram was not representative for panel 2b please remove it also from Fig 2a, otherwise it is misleading to keep only the one that shows more vicinity. A quantification such as the one added in Fig2c is more informative.

Response to comment r4.1b: Histogram removed as suggested.

Comment r4.1.c: the reply is not conclusive.

Response to comment r4.1c: We previously noted that "We've added further detail in the Methods section (Microscopy and image analysis)". This included the following text: "SIM reconstruction was performed after correcting for chromatic aberrations using the channel alignment function in Zen and performing deconvolution using the default settings. 100 nm Tetraspeck beads (Thermo) were adhered to slides and were used to determine channel alignment for each experiment." Thus, we sought to exclude technical artefacts associated with chromatic aberration. Rather than including an 'extended panel of representative figures' we "quantified VEX2 signal diameter in many additional cells and present these data in a violin plot (Fig. 2c)", as detailed in R4.1b.